# Assemblies, synapse clustering, and network topology interact with plasticity to explain structure-function relationships of the cortical connectome

András Ecker[1†], Daniela Egas Santander[1†], Marwan Abdellah[1],
Jorge Blanco Alonso[1], Sirio Bolaños-Puchet[1], Giuseppe Chindemi[1,2],
Dhuruva Priyan Gowri Mariyappan[3,4,5], James B Isbister[1], James King[1],
Pramod Kumbhar[1], Ioannis Magkanaris[1], Eilif B Muller[1,3,4,5], Michael W Reimann[1*]

[1]Blue Brain Project, École polytechnique fédérale de Lausanne (EPFL), Campus Biotec, Geneva, Switzerland; [2]Department of Basic Neurosciences, University of Geneva, Geneva, Switzerland; [3]Department of Neurosciences, Faculty of Medicine, Université de Montréal, Montreal, Canada; [4]Centre de Recherche Azrieli du CHU Sainte-Justine, Montréal, Canada; [5]Mila Quebec AI Institute, Montréal, Canada

**\*For correspondence:**
mwr@reimann.science

[†]These authors contributed equally to this work

## eLife Assessment

This **useful** study presents a biologically realistic, large-scale cortical model of the rat's non-barrel somatosensory cortex, investigating synaptic plasticity of excitatory connections under varying patterns of external activations and characterizing relations between network architecture and plasticity outcomes. The model offers an impressive level of biological detail, addressing many aspects of the cellular and network anatomy and properties, and investigating their relationships to the biologically plausible plasticity. The numerical simulations appear to be well executed and documented, providing an excellent resource to the community. The evidence supporting the main conclusions is **solid** with results being more observational in nature, and minor weaknesses relating to the lack of explanatory power of causal relationships and mechanisms.

**Abstract** Synaptic plasticity underlies the brain's ability to learn and adapt. While experiments in brain slices have revealed mechanisms and protocols for the induction of plasticity between pairs of neurons, how these synaptic changes are coordinated in biological neuronal networks to ensure the emergence of learning remains poorly understood. Simulation and modeling have emerged as important tools to study learning in plastic networks, but have yet to achieve a scale that incorporates realistic network structure, active dendrites, and multi-synapse interactions, key determinants of synaptic plasticity. To rise to this challenge, we endowed an existing large-scale cortical network model, incorporating data-constrained dendritic processing and multi-synaptic connections, with a calcium-based model of functional plasticity that captures the diversity of excitatory connections extrapolated to in vivo-like conditions. This allowed us to study how dendrites and network structure interact with plasticity to shape stimulus representations at the microcircuit level. In our exploratory simulations, plasticity acted sparsely and specifically, firing rates and weight distributions remained stable without additional homeostatic mechanisms. At the circuit level, we found plasticity was driven by co-firing stimulus-evoked functional assemblies, spatial clustering of synapses on dendrites, and the topology of the network connectivity. As a result of the plastic changes, the network became more reliable with more stimulus-specific responses. We confirmed our testable

predictions in the MICrONS datasets, an openly available electron microscopic reconstruction of a large volume of cortical tissue. Our results quantify at a large scale how the dendritic architecture and higher-order structure of cortical microcircuits play a central role in functional plasticity and provide a foundation for elucidating their role in learning.

## Introduction

Learning and memory are orchestrated by synaptic plasticity, the ability of synapses to change their *efficacy* in an activity-dependent manner. Donald O. Hebb's postulate about how synaptic plasticity might manifest was paraphrased with the well-known mantra: '*cells that fire together, wire together*' (**Hebb, 1949**; **Shatz, 1992**). The first proof of coincident pre- and postsynaptic population activity leading to *potentiation* (an increase in efficacy) came from pathway stimulation in hippocampal slices (**Bliss and Lomo, 1973**). It was later confirmed at the neuron pair level (**Markram et al., 1997**; **Bi and Poo, 1998**), and spike-time dependent plasticity (STDP) became a quintessential protocol to study Hebbian plasticity in vitro. In the early 2000s, a plethora of cortical pathways were studied and plasticity proved to be synapse location—and therefore pathway—dependent (**Sjöström and Häusser, 2006**; **Letzkus et al., 2006**; **Froemke et al., 2010**). The molecular substrate of Hebbian coincidence detection is the N-methyl-D-aspartate (NMDA) receptor, which upon removal of the $Mg^{2+}$ block by depolarization, conducts $Ca^{2+}$(**Mayer et al., 1984**). The calcium-control hypothesis, put forward by **Lisman, 1989** postulates that prolonged, moderate amounts of $Ca^{2+}$ lead to *depression* (a decrease in efficacy) while large transients of $Ca^{2+}$ lead to potentiation. By putting these together, it became evident that it is not necessarily the timing of the postsynaptic spike, but rather the depolarization induced-entrance of calcium into the postsynaptic dendrite that is important to evoke changes in synaptic efficacy (**Goldberg et al., 2002**; **Lisman and Spruston, 2005**).

In parallel with slice electrophysiology, Hebbian plasticity was also studied through its effect on behavior via fear conditioning experiments (**McKernan and Shinnick-Gallagher, 1997**) and this line of research led to numerous new techniques for tagging and re-activating cells that participate in newly formed memories (**Tonegawa et al., 2015**). While these studies highlighted the need to study plasticity at the network level, most changes are expected to happen at the synapse level. Therefore, high-throughput methods tracking synaptic proteins like PSD95 (**Ray et al., 2023**) and α-amino-3-hydroxy-5-methyl-4-isoxazolepropionate (AMPA) subunit GluA1 (**Graves et al., 2021**; **Kim et al., 2023**) are currently being developed. While readily applicable to monitor synaptic efficacy in vivo, currently these techniques cannot be supplemented with recordings of neural activity, thus the reason for the changes in efficacy can only be speculated.

The bridge between in vitro pairs of neurons and in vivo behavior is often provided by complementary, simulation-based approaches. Early theoretical work explored the potential link between memories and cells that fire and therefore wire together, concentrating on the storage and retrieval of memories in strongly recurrent networks (**Hopfield, 1982**), which remained an active topic of research (**Fusi and Abbott, 2007**; **Krotov and Hopfield, 2016**; **Widrich et al., 2020**). Inspired by STDP experiments, modelers have postulated diverse *plasticity rules* that fit the most recent experimental findings (**Gerstner et al., 1996**; **Kempter et al., 1999**; **Song et al., 2000**; **Pfister and Gerstner, 2006**; **Clopath et al., 2010**). Models that take into account the crucial role of calcium in driving plasticity outcomes have also been proposed (**Shouval et al., 2002**; **Graupner and Brunel, 2012**; **Rubin et al., 2005**; **Jędrzejewska-Szmek et al., 2017**; **Rodrigues et al., 2023**). The calcium-based model of **Graupner and Brunel, 2012** describes the evolution of intracellular calcium concentration ($[Ca^{2+}]_i$) given the pre- and postsynaptic spike trains and updates the efficacy of the synapse, upon $[Ca^{2+}]_i$ crossing thresholds for depression and potentiation. Bioplausible plasticity rules have been shown to bring about Hebbian *cell assemblies*, i.e., groups of neurons that fire together to enable long-term memory storage (**Litwin-Kumar and Doiron, 2014**; **Zenke et al., 2015**; **Fauth and van Rossum, 2019**; **Kossio et al., 2021**). A common theme in these models is the necessity for homeostatic plasticity to keep the networks stable. While experimental evidence exists for homeostatic plasticity (**Turrigiano and Nelson, 2004**), this has been shown to be too slow to induce stability (**Zenke et al., 2017a**) and alternative mechanism have been proposed (**Vogels et al., 2011**; **Delattre et al., 2015**). While these studies provide mechanistic explanations of learning and memory, they used point-neuron models, therefore neglecting the structural and functional importance of dendrites in plasticity (with

the exception of *Bono et al., 2017*; *Kastellakis and Poirazi, 2019*). The compartmentalized nature of dendritic trees gives rise to spatial clustering of synapses (*Kastellakis and Poirazi, 2019*; *Farinella et al., 2014*; *Iacaruso et al., 2017*; *Tazerart et al., 2020*) and local, non-linear voltage events (*Poirazi et al., 2003*; *Stuart and Spruston, 2015*) both of which are thought to contribute to the removal of the $Mg^{2+}$ block from NMDA receptors and therefore gate plasticity.

To investigate the effect of dendrites and multi-synaptic interactions for plasticity at the network level, in this study we equipped the biophysically detailed, large-scale cortical network model of *Isbister et al., 2023* with our recently developed model of functional plasticity (*Chindemi et al., 2022*) between excitatory cells (*Figure 1*). At this level of detail, a calcium-based plasticity model is the natural choice since calcium dynamics are already part of the simulation. Moreover, it allowed us to model in vivo-like conditions, specifically the low extracellular calcium concentration ($[Ca^{2+}]_o$, *Chindemi et al., 2022*), which has been experimentally shown to reduce plasticity (*Inglebert et al., 2020*, *Figure 1E*). As we had access to the pre- and postsynaptic activity and efficacy of millions of synapses, including their dendritic location, we could characterize the rules governing plasticity at the microcircuit level. During simulated evoked activity, plasticity acted sparsely and specifically, although it was still able to reorganize the network's dynamics, manifesting in more pattern-specificity after plasticity. Potentiation dominated in amplitude and depression counteracted it in frequency, which led to stable firing rates without explicitly introducing any homeostatic terms (*Zenke et al., 2017a*; *Turrigiano and Nelson, 2004*). We found plasticity was driven by the network topology beyond pairwise connectivity. Specifically, by co-firing cell assemblies, spatial clustering of synapses on dendrites, and the centrality of a connection in the network. Furthermore, we confirmed our testable predictions in the (*MICrONS, 2021*) dataset, an openly available electron microscopic reconstruction.

## Results

To study how plasticity and various network level features interact to explain structure-function relationships, we used a bio-realistic, large-scale cortical model of the rat non-barrel somatosensory cortex (nbS1). By doing so, we had a continuous readout of both the activity of all neurons and the efficacy of the synapses between them under in vivo-like, low $[Ca^{2+}]_o$ conditions (~1 mM instead of 2-2.5 mM in vitro). The current model improves on *Markram et al., 2015* in terms of both anatomical, e.g., atlas-based cell composition and placement (described in *Reimann et al., 2024a*), and physiological properties, e.g., improved single-cell models, multi-vesicular synaptic release, and layer-wise compensation for missing synapses (described in *Isbister et al., 2023*). For this study, we used a seven column subvolume comprising 211,712 neurons in 2.4 mm$^3$ of tissue (*Figure 1A*). In line with the biological variability, excitatory cells are modeled as a diverse set of morphologies (*Reimann et al., 2024a*; *Kanari et al., 2019*; *Figure 1B*) equipped with conductances distributed across all electrical compartments (*Reva et al., 2023*; *Figure 1—figure supplement 1A*). The connectivity and synaptic physiology of these cells were extensively validated (*Isbister et al., 2023*; *Reimann et al., 2024a*; *Figure 1C*; *Figure 1—figure supplement 1C*). To deliver input with spatio-temporal precision to the circuit, we used thalamic fibers from the ventral posteriomedial nucleus of the thalamus (VPM) and the high-order posteriomedial nucleus of the thalamus (POm; *Figure 1D*; *Meyer et al., 2010*).

To simulate long-term plasticity, we integrated our recently published calcium-based plasticity model that was used to describe functional long-term potentiation and depression between pairs of pyramidal cells (PCs) (*Chindemi et al., 2022*). In short, the model follows the formalism of *Graupner and Brunel, 2012*, where pre- and postsynaptic spikes led to changes in synaptic $[Ca^{2+}]_i$ (*Figure 1E*). Calcium entering through NMDA receptors and voltage-dependent calcium channels (VDCCs) contributes to $[Ca^{2+}]_i$ (*Equation 2* in Methods). When the integrated calcium trace of a synapse crosses the threshold for depression ($\theta_d$) or the higher one for potentiation ($\theta_p$), *synaptic efficacy* ($\rho$) is updated (*Figure 1E* left; *Equation (1)* in Methods). Changes in $\rho$ are then converted into changes in the utilization of synaptic efficacy ($U_{SE}$), a variable of the Tsodyks-Markram model of short-term plasticity describing the baseline probability of vesicle release (*Tsodyks and Markram, 1997*), and the peak AMPA receptor conductance ($\hat{g}_{AMPA}$; *Equations (7) and (8)* in Methods, respectively). As a result of updating $U_{SE}$, short- and long-term plasticity are tightly coupled in the model (*Markram and Tsodyks, 1996*; *Costa et al., 2015*; *Deperrois and Graupner, 2020*). In the absence of $[Ca^{2+}]_i$ influx $\rho$ exhibits bistable dynamics (*Lisman, 1985*; *Graupner and Brunel, 2012*) and at initialization, synapses are

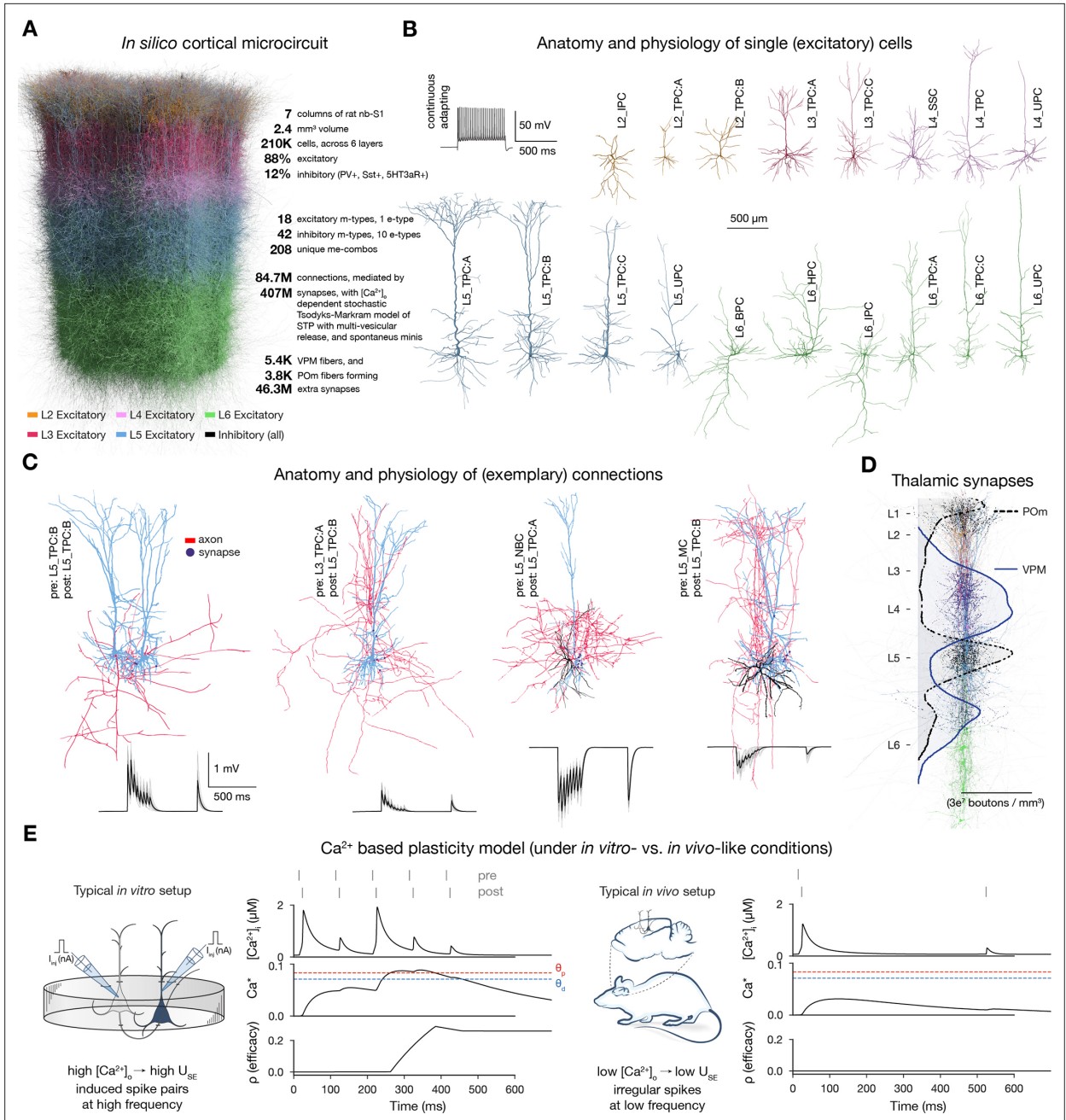

**Figure 1.** Overview of the cortical network model and calcium-based plasticity rule. (**A**) Rendering of the seven column subvolume of rat nbS1. (10% of the cells are shown). (**B**) Representative morphologies for the 18 excitatory morphological types (m-types) and their typical firing pattern (top left). Neurite diameters are scaled (2x) for better resolution. (**C**) Exemplary connections to layer 5 (thick-tufted pyramidal cells, L5_TPC:A and L5_TPC:B) L5 TTPCs (top) and their short-term dynamics (bottom). Neurite diameters are uniform for better visibility. On the bottom: thin gray lines represent 20 individual trials, while the thicker black ones their means. (**D**) Bouton density profiles of thalamocortical fibers, and locations of ventral posteriomedial nucleus of the thalamus (VPM) (black) and posteriomedial nucleus of the thalamus (POm) (purple) synapses on neurons (in a 5 μm radius subvolume). The spatial scale bar on B applies to the whole figure. Panels adapted from *Isbister et al., 2023*; *Chindemi et al., 2022*; *Reimann et al., 2024a*. (**E**) Main variables of the plasticity model during coincident activation of the pre- and postsynaptic neurons for an exemplary synapse. Left: under in vitro-like conditions (adapted from *Chindemi et al., 2022*). Right: same pair of neurons under in vivo-like conditions. Schematics on their left illustrate the difference between typical in vitro and in vivo protocols.

The online version of this article includes the following figure supplement(s) for figure 1:

**Figure supplement 1.** Physiology of excitatory cells and E to E connections.

**Figure supplement 2.** Spike-time dependent plasticity (STDP) and synapse-specific parameters of the plasticity model.

*Figure 1 continued on next page*

*Figure 1 continued*

**Figure supplement 3.** Calibration of the in vivo-like network state.

**Figure supplement 4.** Activity of the thalamic fibers.

assumed to be at one of the two fixed points (fully depressed ones at $\rho = 0$ and fully potentiated ones at $\rho = 1$) and their assignment to these states is pathway-specific (*Figure 1—figure supplement 1C3*).

We calibrated layer-wise spontaneous firing rates and evoked activity to brief VPM inputs matching in vivo data from *Reyes-Puerta et al., 2015* (*Figure 1—figure supplement 3*). Spontaneous activity was driven by somatic injection of a layer- and cell-type-specific noisy conductance signal (*Isbister et al., 2023*, see Methods). Evoked activity was driven by a thalamocortical input stream already described in *Ecker et al., 2024*. In short, 10 VPM input patterns were repeatedly presented in random order with a 500 ms inter-stimulus interval, together with a non-specific POm input. These VPM patterns were defined with varying degrees of overlap in the sets of activated fibers (*Figure 2A*, *Figure 1—figure supplement 4*; see Methods). An exemplary raster plot, characterizing the evoked state of the plastic network is shown in *Figure 2B*.

## Plasticity changes promote stable network activity

In vivo, plasticity does not seem to create macroscopic network instability. Therefore, it is a critical first step to validate our model in this regard. Previous theoretical work has shown that while embedding simple STDP rules in spiking networks without fast homeostatic plasticity leads to exploding firing rates (*Morrison et al., 2007*; *Zenke et al., 2017a*), this is not necessarily the case for calcium-based rules (*Higgins et al., 2014*; *Graupner et al., 2016*; *Chindemi, 2018*; *Wang and Aljadeff, 2022*). To assess the stability of our network while undergoing plasticity with our specific choice of calcium-based rule, we simulated 10 min of biological time of in vivo-like, stimulus-evoked network activity and verified that excitatory firing rates remained stable throughout the entire length of the simulation (*Figure 2C*).

When comparing the amount of changes in $\rho$ across time steps, we found that most of the plastic changes happened in the first 1–2 min of the simulation, after which they stabilized (*Figure 2D1*). While small changes were still apparent towards the end of the simulation, by visualizing individual synaptic efficacy traces, we confirmed that most of them oscillated around a dynamic fixed-point and the amount of changes in the second half of the simulation were negligible (*Chindemi, 2018*, *Figure 2—figure supplement 1*). The results were similar (*Figure 2E1*, black line) when we averaged $\rho$ values within a connection, i.e., all synapses between a pair of neurons (on average 4.1 ± 2.3 synapses per connection; *Figure 1—figure supplement 1C1*). Considering changes at the connection level allowed us to compare our results against a traditional spike pair-based STDP rule (*Gerstner et al., 1996*; *Kempter et al., 1999*; *Song et al., 2000*, see Methods). As it is evident from previous theoretical work that simply equipping networks with pair-based STDP rules leads to exploding firing rates due to a positive feedback loop (*Morrison et al., 2007*; *Zenke et al., 2017a*), we turned to a stricter control and executed the STDP rule on the 35 M excitatory spikes from our simulation. We observed that the STDP rule kept inducing the same magnitude of changes throughout all 10 min of the simulation instead of stabilizing after a transient phase (*Figure 2E*, pink line). To further assess whether the remaining changes in mean $\rho$ were in the form of fluctuations around a dynamic fixed-point, we compared the changes in the second half of the simulation to a random walk with the same step size (see Methods). The magnitude of changes were well below the random walk control for our simulation, while the changes from the STDP rule were above both, but below a random walk with a step size fitted to the changes induced by the STDP (*Figure 2E2*, see Methods). Consequently, the lifetime of synaptic efficacy configurations in the network are longer when using a calcium-based rule than an STPD rule or a random walk. This demonstrates that not only the scale of the simulated network, its biorealistic connection probabilities and the low, in vivo-like rates contribute to the stability of changes observed in our simulation, but also the use of a calcium-based plasticity model (*Chindemi et al., 2022*) in line with previous modeling insights (*Higgins et al., 2014*).

Next, we went beyond keeping a stable network stable and investigated if an unstable network exhibiting synchronous activity (*Figure 2F*, left) could be brought back to a stable, asynchronous regime by the calcium-based plasticity rule. The synchronized activity state was the result of simulating

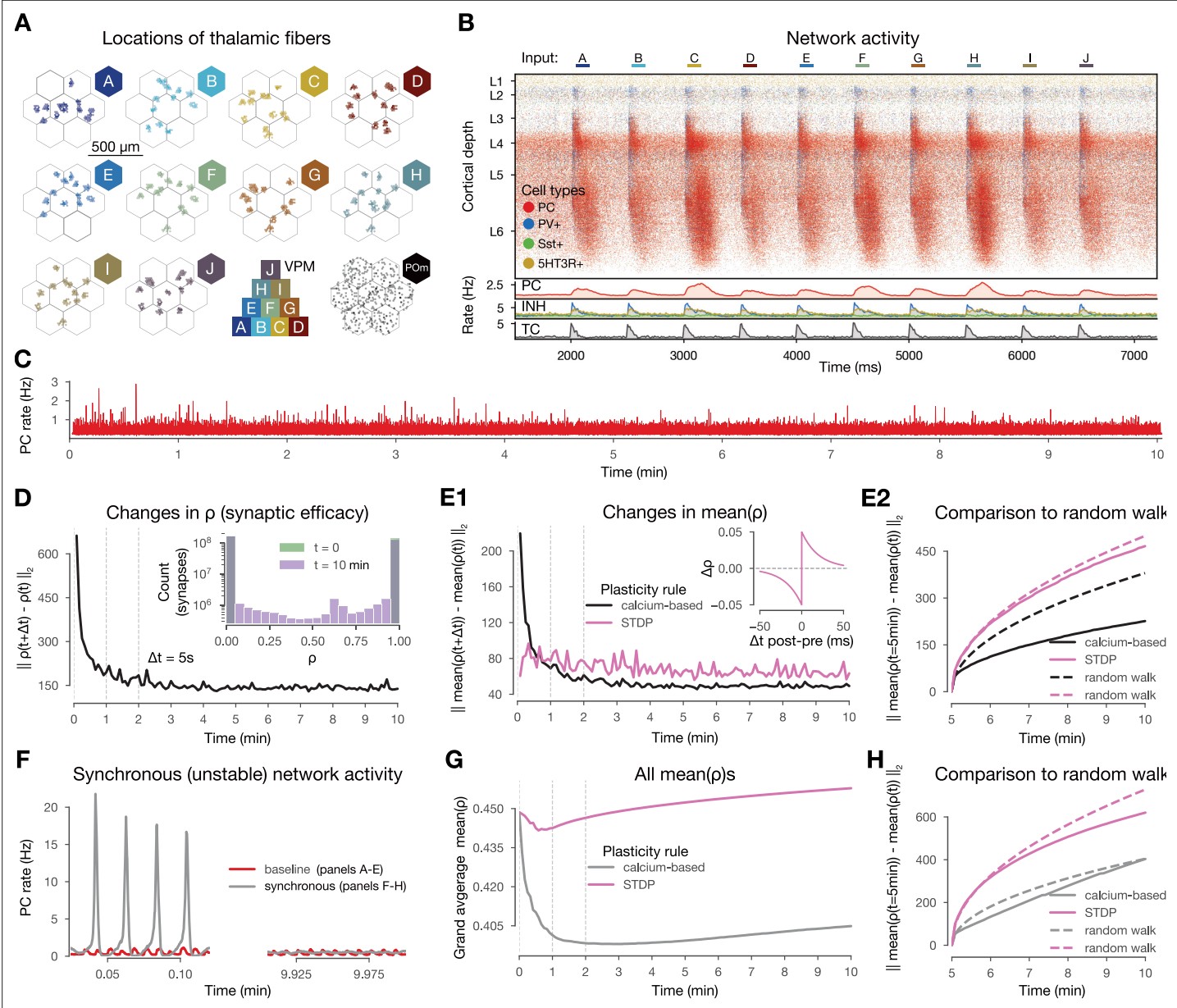

**Figure 2.** Changes in synaptic efficacy stabilize and they promote macroscale stable dynamics. (**A**) Top-down view of the spatial arrangement of the ventral posteriomedial nucleus of the thalamus (VPM) fiber centers associated with the 10 input patterns. Bottom row third: pyramid-like overlap setup of VPM patterns, fourth (bottom right) posteriomedial nucleus of the thalamus (POm) fibers (high-order thalamic input) associated with all stimuli. (**B**) Raster plot of the microcircuit's activity and the population firing rates below. The y-axis shows cortical depth. As cortical layers do not have the same cell density, the visually densest layer is not necessarily the most active. Panel adapted from *Ecker et al., 2024*. (**C**) Firing rate of excitatory cells during the 10-min-long simulation. (**D**) Plastic evolution of synaptic efficacy ($\rho$): L2 norm of changes in $\rho$ across time. Inset shows the distribution of $\rho$ values in the beginning (green; strickly 0 s and 1 s, see Methods) and end (purple) of the simulation. (**E**) Evolution of mean $\rho$ (aggregated over multiple synapses within a single conncnnection**E1**) L2 norm of changes in mean $\rho$ (black) across time against STDP control in pink (inset, see Methods). (**E2**) L2 norm of changes in $\rho$ (similar to **E1**, but) compared to $\rho_{t=5\text{ minutes}}$ in black, spike-time dependent plasticity (STDP) in pink, and their random walk controls (see Methods) with the same colors but dashed lines. (**F**) Firing rates of excitatory cells from a simulation with higher $[Ca^{2+}]_o$ and thus population bursts (*Figure 2—figure supplement 2*, *Markram et al., 2015* Methods) in gray and the baseline one in red as on panel C. (**G**) Grand average mean $\rho$ values across time for the synchronous simulation in gray and for the corresponding STDP control in pink. (**H**) Same as E2 but for the synchronous simulation (calcium-based plasticity rule in gray instead of black).

The online version of this article includes the following figure supplement(s) for figure 2:

**Figure supplement 1.** Changes in synaptic efficacy during plasticity.

**Figure supplement 2.** Synchronous (unstable) network activity.

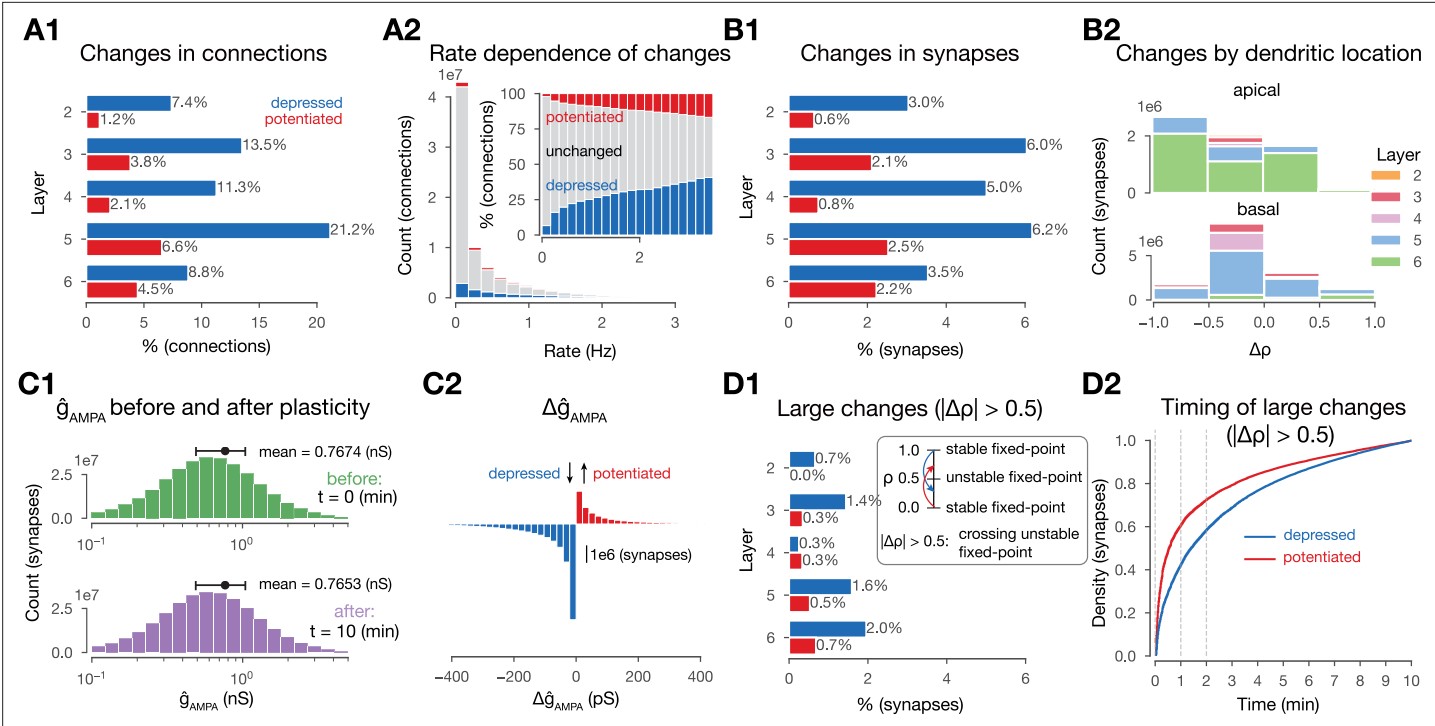

**Figure 3.** Changes in synaptic efficacy are sparse. (**A1**) Propensity of changes in connections (mean $\rho$ values) across layers. Connections are grouped into layers based on the soma location of the postsynaptic cell. Percentages are calculated within layers, i.e., 7.4% depression (in blue) for layer 2 (L2) means that among all connections that have the postsynaptic cell in L2, 7.4% depressed. (**A2**) Plastic changes in mean $\rho$ vs. mean pairwise firing rates of the pre- and postsynaptic cell of a given connection. (**B1**) Same as A1 but for $\rho$ (changes in synapses rather than connections). (**B2**) Layer- and neurite type-wise (apical and basal dendrites) distribution of $\Delta\rho$. Only synapses that underwent any plastic change (~15 M) are shown. (**C1**) Distribution of $\hat{g}_{AMPA}$ in the beginning (in green) and end (in purple) of the simulation. (**C2**) Plastic changes in $\hat{g}_{AMPA}$ leading to the slight shift in the distribution on **C1**. (**D1**) Same as B1, i.e., propensity of changes at the synapse level, but for synapses that cross the $\rho_* = 0.5$ unstable fixed-point. D2: Cumulative histogram of $\rho$ crossing the unstable fixed-point against time for depressing synapses in blue and potentiating ones in red.

The online version of this article includes the following figure supplement(s) for figure 3:

**Figure supplement 1.** Changing connections in plastic control simulations.

a higher $[Ca^{2+}]_o$ and thus higher release probability ($U_{SE}$, see Methods). Surprisingly, the firing rate flipped from synchronous to asynchronous (*Figure 2F*, right) and it did so within a minute of biological time (*Figure 2—figure supplement 2B*). This was likely driven by a sudden decrease in the mean $\rho$ values over synapses in the network, which decrease could not be observed for the STDP rule (*Figure 2G*). As before, both rules led to smaller changes than their corresponding random walk controls (*Figure 2H*). Taken together, our results indicate that the calcium-based rule evolves an over-excitable network towards lower excitability, which in this case is a role that a homeostatic rule would play (*Turrigiano and Nelson, 2004*; *Zenke et al., 2017a*).

In summary, the calcium-based plasticity rule promotes stability in our network. Indeed, when applied to asynchronous activity, it generates changes in synaptic efficacy that stabilize after a transient phase without leading to exploding firing rates. Furthermore, when applied to an unstable synchronous activity regime, the mean synaptic efficacy of the whole network consistently decreases making the network less excitable and fosters stability in the macroscopic activity.

## Plasticity induces sparse and specific changes driven by the stimulus and the network's topology

We have shown that the changes in synaptic efficacy stabilize after a transient period, which raises the question: What predictions does the model make about the structure of the plastic changes in vivo? First of all, the changes were sparse, implying that the configuration of synaptic efficacy of the overall network remained largely unchanged. Indeed, we found that the number of connections changing

was below 18% across the whole network and remained low when restricted to individual layers (*Figure 3A1*). Moreover, the propensity of changes increased as the pairwise firing rates increased (*Figure 3A2*), in line with previous modeling insights (*Litwin-Kumar and Doiron, 2014*; *Graupner et al., 2016*). On the level of individual synapses changes were even sparser (*Figure 3B1*), and in both cases, depression was more common than potentiation. Layer 5 (L5) PCs contributed mostly to changes on the basal dendrites, while apical changes happened mostly on L6 PCs (*Figure 3B2*).

So far, we characterized plastic change in terms of $\rho$, as this parameter is bounded to the $[0, 1]$ interval and thus easy to interpret across pathways. Another parameter affecting the function of the synaptic connection is $\hat{g}_{AMPA}$. We found a minimal decrease (–2 pS, *Figure 3C1*) in its mean value due to plasticity, which was explained by frequent depression and a heavy tail of potentiation amplitudes (*Figure 3C2*). The distribution of $\hat{g}_{AMPA}$ remained lognormal, in line with biology (*Buzsáki and Mizuseki, 2014*; *Rößler et al., 2023*, *Figure 3C1*, bottom). The plasticity model employed was bistable around $\rho_* = 0.5$, i.e., in the absence of further $Ca^{2+}$ influx, values above 0.5 would converge towards 1 and values below towards 0 (see *Graupner and Brunel, 2012* and Methods). The fraction of changes in $\rho$ that crossed this threshold was slightly higher for depression (26%) than for potentiation (23%) (compare *Figure 3D1-B1*). Furthermore, the vast majority of these crossings occurred early, with 60% of potentiation and 40% of depression happening within the first minute (*Figure 3D2*).

Additionally, we hypothesized that both the amount, i.e., the level of sparsity, and the structure of plastic changes are non-random and instead shaped by the stimuli and the underlying network's topology. We confirmed our hypothesis first for the amount of plastic changes. We ran control simulations where we delivered random Poisson spikes on the same VPM fibers with the same mean rate, but without the spatio-temporal structure of the stimuli. We found that this reduced the number of connections undergoing plasticity by 25%, demonstrating the importance of stimulus structure over simple firing of pre- and postsynaptic neurons (*Figure 3—figure supplement 1*). Additionally, without synaptic transmission, plasticity was reduced even further: In simulations of stimulus streams where intrinsic connectivity between the simulated neurons was cut, we found occasional changes in $\rho$, but the number of changing synapses was an order of magnitude below baseline conditions (*Figure 3— figure supplement 1*). Thus, our plasticity model is not strictly Hebbian, since the effect of postsynaptic firing alone could change synaptic efficacy, although presynaptic release was required for most of the observed changes (*Graupner and Brunel, 2012*; *Graupner et al., 2016*). When the external stimuli were also left out, negligible plastic changes occurred (*Figure 3—figure supplement 1*).

We then confirmed that the network's topology-shaped plastic changes as well. In particular, we found layer-to-layer pathway specificity in the amount of plastic changes observed. That is, the amount of changes between layers differed from those expected in a random control with the same pre- and postsynaptic populations (*Figure 4A2* vs. *Figure 4A3*). We further quantified the difference in the subnetwork of changing connections from the random control by counting *directed simplices* (*Figure 4B*). These are motifs that have previously shown to be linked to network function (*Reimann et al., 2017*) as well as quantify the complexity of the network's topology (*Kahle, 2009*; *Bobrowski and Kahle, 2018*). Succinctly, a directed simplex of dimension $k$, is a motif on $k + 1$ neurons with all-to-all feed-forward connectivity (*Figure 4A1* right, see Methods). We found strong overexpression of simplices in the subgraph of changing connections compared to the control (*Figure 4B*). Furthermore, the maximal simplex dimension found in the subgraph exceeded the value expected by chance by two. This result shows that the connections changing are not determined only by their pre- and postsynaptic populations but also depend on their embedding in the whole network. We will explore this further in the Network-level metrics predict plasticity and separate potentiation from depression section.

Finally, we showed that the input stimuli shaped plastic changes by running 2-min-long plastic simulations in which we only presented a single pattern (several times, with the same 500 ms inter-stimulus interval as before) and compared the changes in mean $\rho$ matrices. By using an input - output distance correlation analysis, we found that the distances between these mean $\rho$ matrices were highly correlated with the input distances of the associated VPM fiber locations ($r = 0.716$, $p < 0.0001$, *Figure 4C2*, *Figure 4—figure supplement 1A*). Furthermore, this required using Euclidean distance between the $\rho$ matrices on the output side, implying that both the mean $\rho$ values themselves and their position in the network are relevant. When either Hamming distance (taking only the identity of changing connection into account) or Earth mover's distance (taking only the distribution of mean $\rho$

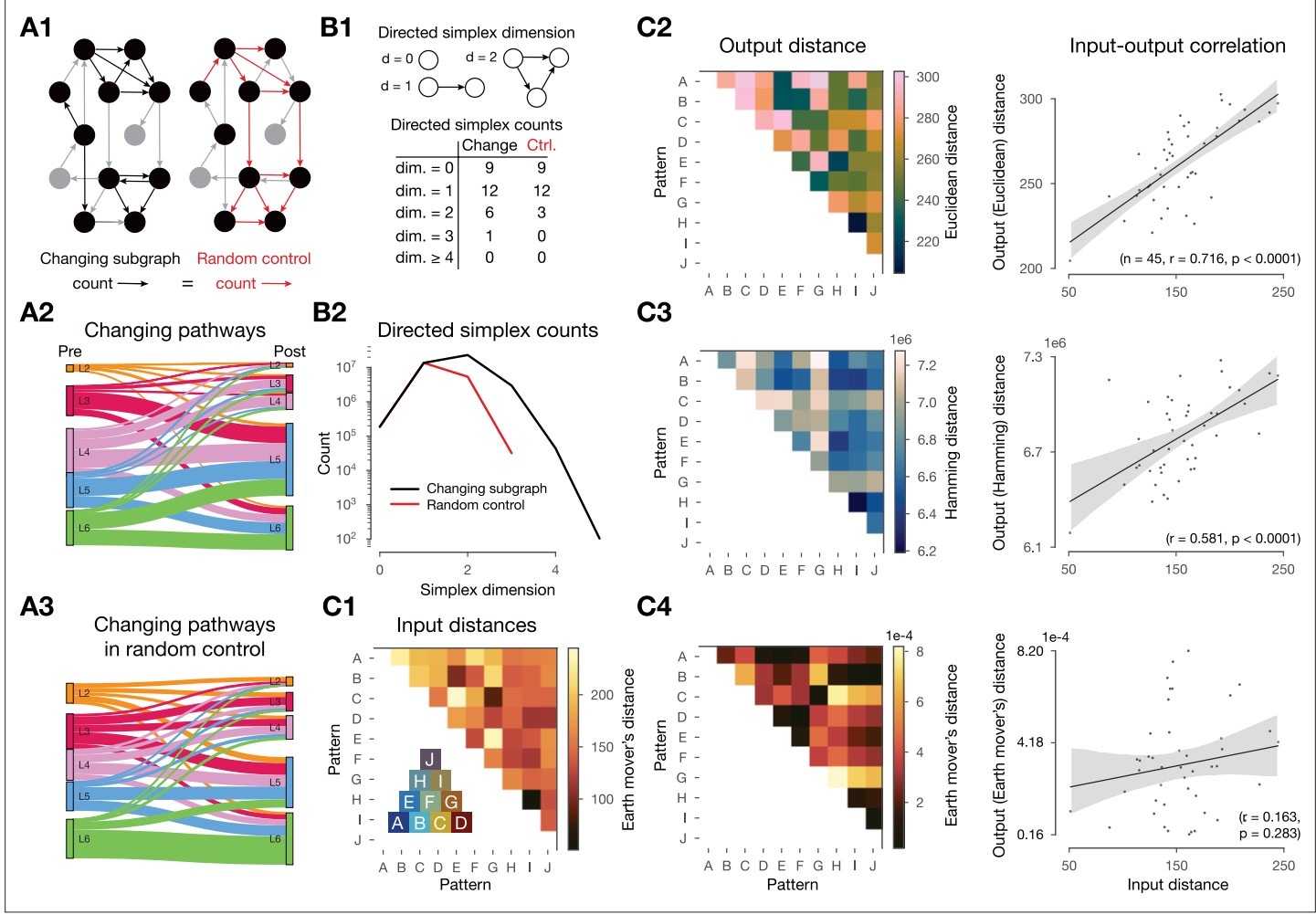

**Figure 4.** Changes in synaptic efficacy are non-random and shaped by the networks topology and the input stimuli. (**A**) Layer-wise distribution of changing connections. (**A1**) Left: schematic of the *changing subgraph.* In black, connections whose efficacies are changing and their pre- and postsynaptic populations. In gray, connections whose efficacies do not change and neurons not partaking in a changing connection. Right: random control of the changing subgraph, generated by randomly selecting the same number of connections as the changing connections (red edges) between the same pre- and postsynaptic populations (black nodes). (**A2**) Sankey plot of layer-wise distribution of changing connections. Thickness of lines is proportional to the number of changing connections between pre- and post node populations. (**A3**) As A2, but for the random control (A1 right). (**B**) Directed simplex counts in the changing subgraph and its random control. (**B1**) Schematic of directed *k*-simplices and their counts in the schematic graphs on A1. Note that by construction the control must have the same number of 0- and 1-simplices (see Methods) which correspond to the number of cells and connections in the subnetwork. B2: Directed simplex counts across dimensions in the changing subgraph (black) and its random control (red). (**C**) Input - output distance correlations. (**C1**) Input distances as the Earth mover's distance of the ventral posteriomedial nucleus of the thalamus (VPM) fiber locations (see *Figure 2A*). Inset shows the overlap (based on Hamming distance, see Methods) of pattern fibers. (**C2**) Output distances as Euclidean distance of mean $\rho$ matrices in the 2-min-long single pattern simulations. To its right: correlation of input and output distances. (**C3**) Same as C2 but with Hamming distance. (**C4**) Same as C2 but with Earth mover's distance (on the output side as well).

The online version of this article includes the following figure supplement(s) for figure 4:

**Figure supplement 1.** Analysis results across three repetitions of the same simulations.

values into account) were used instead, we found weaker and non-significant correlations, respectively (*Figure 4C3 and C4*, respectively).

In summary, we observed that ~7% of synapses undergo long-term plasticity under realistic in vivo-like conditions in 10 min of biological time, and most of these synapses are on L5 PC's basal dendrites. Moreover, the changes are not random, but depend not only on the firing rates but also on the recurrent connectivity and input stimuli. Potentiation dominated in amplitude, while depression counteracted it in frequency, keeping the network stable without needing to model homeostatic plasticity (*Zenke et al., 2017a*; *Turrigiano and Nelson, 2004*).

## More frequent plastic changes within and across cell assemblies

We have shown that plastic changes are sparse and highly specific. We, therefore, tried to understand the rules determining which synapses changed. From the parametrization of the plasticity model, we learned that presynaptic spikes contribute orders of magnitude higher $[Ca^{2+}]_i$ than postsynaptic ones if the NMDA receptors are fully unblocked (*Figure 1—figure supplement 2B*). Thus, in order to effectively depolarize the dendrites and unblock NMDA receptors, spikes at low, in vivo-like rates must be synchronized in time. Therefore, we specifically hypothesized that plasticity of connections may be structured by the membership of participating neurons in Hebbian cell assemblies (*Hebb, 1949*; *Harris, 2005*). In our previous analysis (*Ecker et al., 2024*), and in line with experimental results (*Harris, 2005*; *Song et al., 2005*; *Perin et al., 2011*), we observed that the number of afferents from an assembly is a great predictor of a neuron's membership in an assembly. Strong positive interactions were also found *across assemblies*, but only when the direction of innervation reflected the temporal order of assembly activation. These results, combined with the biophysics of the plasticity model, suggest that connections within an assembly and the ones between temporally ordered assemblies, are expected to undergo plastic changes with a higher probability.

To test this hypothesis, we detected cell assemblies, based on their co-firing function, from the in silico spiking activity of the 10-min-long plastic simulation using methods established by experimentalists (*Carrillo-Reid et al., 2015*; *Herzog et al., 2021*, see Methods). In short, spikes were binned and bins with significantly high firing rates were hierarchically clustered (*Figure 5A*). These clusters correspond to the functional assemblies of neurons (*Figure 5C*), with a neuron being considered a member if its spiking activity correlates with the activity of an assembly significantly stronger than chance level. Since time bins and not neurons, were clustered in the first place, this method yields one assembly per time bin and single neurons can be part of several assemblies (*Figure 5B and D*). To foster effective discussions in the field (*Miehl et al., 2023*) urged to call co-firing neurons, detected from purely functional activity *ensembles*, and strongly interconnected ones - usually by design in modeling studies - *assemblies*. While we agree with the distinction, we prefer to call the groups of neurons used for analysis in the study *functional assemblies* as a mix of the two terms, since we have shown in our previous investigation using the same circuit model that they not only co-fire but are interconnected more than expected (*Ecker et al., 2024*). Note, that, in contrast to classical modeling studies, where assemblies are usually defined based on their strong internal connectivity (i.e. their structure *Litwin-Kumar and Doiron, 2014*; *Zenke et al., 2015*; *Fauth and van Rossum, 2019*; *Kossio et al., 2021*) our functional assemblies are detected from the activity of the network only, eliminating a potential bias arising from links between structural properties of changing synapses and the assembly detection method.

Functional assemblies were activated in all stimulus repetitions and a series of three to four assemblies remained active for 190 ± 30 ms (*Figure 5B*). Pattern C elicited the strongest response, while pattern H the weakest, and the responses of patterns H and I were the most similar to each other, as expected, since they share 66% of the VPM fibers (*Figure 2A*). Functional assembly activations had a well-preserved temporal order - with some of them always appearing early during a stimulus, while others later - and from now on we will refer to them as *early*, *middle*, and *late assemblies*, and will order them in the figures accordingly.

When checking the propensity of changes within and across functional assemblies, we indeed found more synapses undergoing long-term plasticity (*Figure 5F*). While only 5% of synapses depressed in the whole dataset, we found up to 13.8% when restricting the analysis to assemblies. Similarly, compared to 2% of all synapses potentiating, we observed up to 5.2% when restricting to functional assemblies. Interestingly, large values were found in the off-diagonal entries (*Figure 5F*), i.e., synapses across assemblies underwent more plastic changes than the synapses within these assemblies. In our model, the initial $\rho$ values are pathway-specific and highest in L4 pathways (*Brémaud et al., 2007*, *Figure 1—figure supplement 1C3*). Therefore, early assemblies, with large numbers of L4 cells have a higher than average initial $\rho$ (*Figure 5C and E*, respectively), thus their synapses are more likely to depress (*Figure 5F* left). As early assemblies are stimulus-specific, and thus not part of the same assembly sequences, synaptic depression between these cells can be seen as some kind of stimulus separation. On the other hand, late assemblies, that are predominantly composed of cells from the deep layers, have a low initial $\rho$ (*Figure 5E*; *Figure 1—figure supplement 1C3*) and synapses towards them are more likely to potentiate (*Figure 5F* right). These functional assemblies are mostly

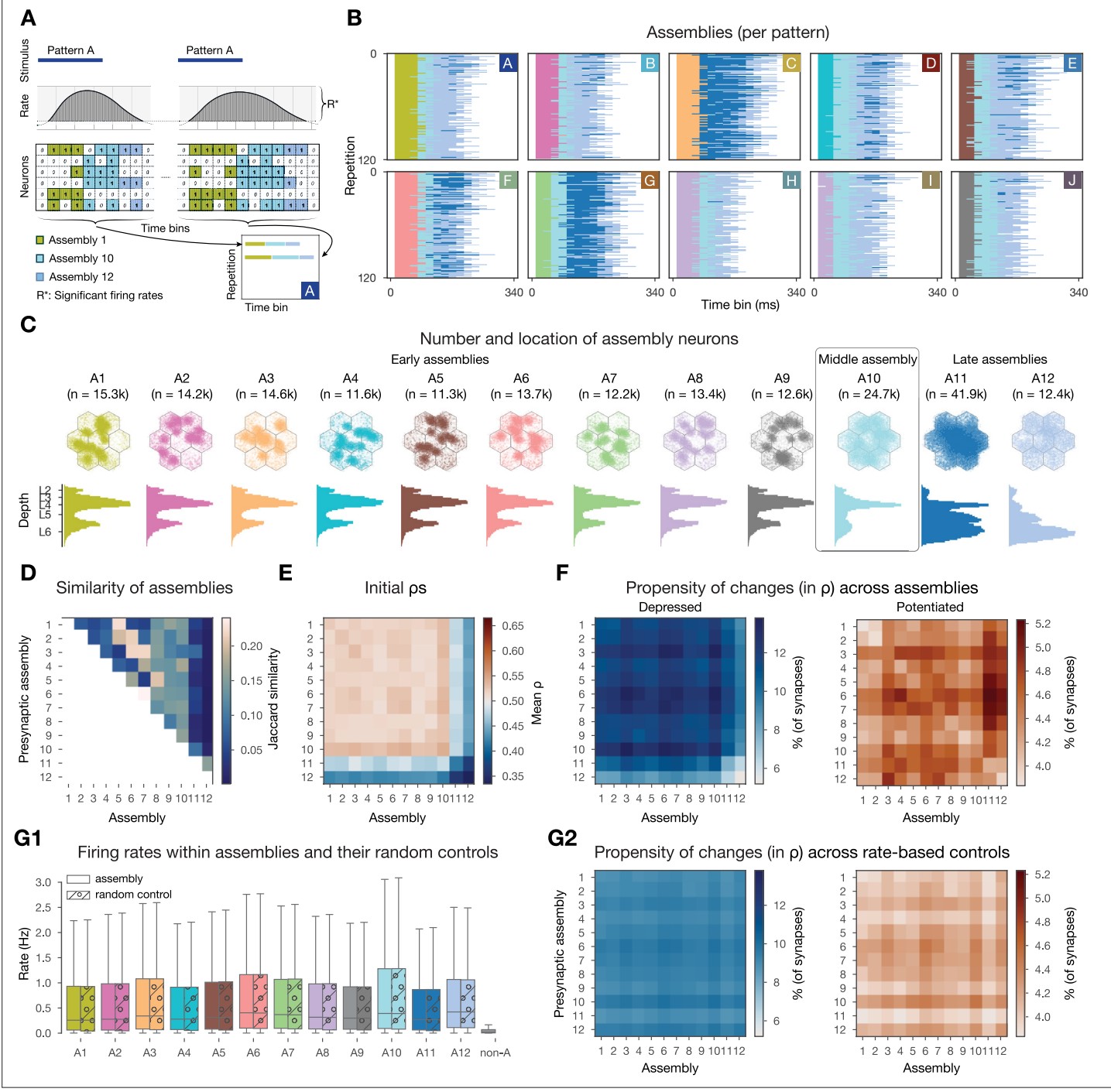

**Figure 5.** Changes are more frequent within and across cell assemblies. (**A**) Schematics of the assembly detection pipeline. Significant time bins are clustered by cosine similarity and these clusters define functional assemblies: olive, light blue, steel blue (see Methods). Bottom right, visual summary of the organization of pattern-specific assembly sequences of the two repetitions of pattern A. (**B**) Activation of functional assembly sequences for multiple repetitions of the patterns presented (as in A bottom right). Each row within the 10 matrices corresponds to a single repetition of a given pattern. White: non-significant time bins. (**C**) Number and location of neurons in each cell assembly. Constituent neurons are those for which the correlation of their spiking activity within the corresponding time bins is significantly stronger than chance (see Methods). Top: top-down view, bottom: depth-profile. (**D**) Jaccard similarity (intersection over union) between cell assemblies. (**E**) Initial mean $\rho$ of within- and cross-assembly synapses. (**F**) Propensity of depression and potentiation of within- and cross-assembly synapses in blue on the left and in red on the right, respectively. Since assemblies are overlapping (see **D**) some synapses are counted in multiple pre- and postsynaptic assembly pairings. (**G**) Firing rate-based random controls of changes within and across cell assemblies. (**G1**) Firing rate distributions of cell assemblies and their corresponding firing rate matching random controls. Last

*Figure 5 continued on next page*

*Figure 5 continued*

gray box depicts rates of non-assembly (non-A) neurons. (**G2**) Same as F above, but for rate-based controls of assemblies. Color bars are the same as in F for easier comparison.

The online version of this article includes the following figure supplement(s) for figure 5:

**Figure supplement 1.** Rate-based controls.

non-specific and participate in all assembly sequences, thus the potentiation of their efferents means a general strengthening of the stimulus response as a whole.

We have shown above that connections between pairs of neurons with high firing rates are more likely to undergo plastic changes (*Figure 3A2*). Thus, we were wondering if firing rate alone explains the increased propensity of changes we observed when restricting our analysis to assembly synapses. To this end, for each assembly, we randomly sampled the same number of neurons with the same firing rate distribution from the whole circuit (*Figure 5G1*) and studied the changes of their synapses. We observed fewer synapses undergoing plastic changes and also a loss of specificity in these rate-based controls (*Figure 5G2*, *Figure 5—figure supplement 1A*). In conclusion, co-firing in a Hebbian cell assembly is a better predictor of plastic changes than just firing at a high rate.

Together, these results indicate that, in line with 70-y-old predictions, cells that fire together wire together (*Hebb, 1949*). Our contribution lies in making the qualitative statement above into a quantitative one: Under in vivo-like conditions, cells that fire together more than expected have three times higher chances of changing the efficacy of their synapses.

## Synapse clustering interacts with assembly formation and plastic changes in complex ways

In addition to co-firing, a group of presynaptic neurons is more effective in depolarizing a given dendritic branch if they all innervate the same branch, i.e., they form a spatial synapse cluster (*Kastellakis and Poirazi, 2019*; *Farinella et al., 2014*; *Iacaruso et al., 2017*; *Tazerart et al., 2020*; *Kirchner and Gjorgjieva, 2021*; *Kirchner and Gjorgjieva, 2022*). To quantify this, we selected the 10 most innervated L5 TTPCs (thick-tufted pyramidal cells) within a cell assembly and then detected spatial clusters of synapses, defined as at least 10 synapses within a 20 μm stretch of a single dendritic branch (Methods; ~15k clustered synapses per assembly on average). Next, we grouped all synapses on these 10 selected neurons per assembly into four categories based on assembly membership of the presynaptic neuron and whether the synapse was part of a cluster or not (see exemplary clustered assembly synapses in *Figure 6A*). While some of the $\rho$ values of clustered synapses within an assembly, underwent small constant changes, many of them changed at the same time (vertical stripes on *Figure 6B*); indicating the importance of spatial co-localization for plasticity.

To systematically quantify this effect, we calculated the likelihood of plastic change in each category: (not) in an assembly and (non-) clustered, by contrasting the conditional probability of observing it in a given category with the probability of observing any change irrespective of the category (see *Equation 15* in Methods; *Figure 6C*). Surprisingly, clustered synapses within an assembly were only likely to undergo plastic changes in the larger, but less general late assembly, A11, but not in early and middle assemblies (*Figure 6C* black arrows top). In fact, the only category that was likely to change across all assemblies is that of non-clustered within-assembly synapses. Therefore, we concluded that synapses within the other three categories tend to keep their initial arrangement of $\rho$ values. To study the initial arrangements as well, we repeated the same analysis on the probability of the initial value of $\rho$ to be either 0 or 1. Initial $\rho$ values in early and middle assembly synapses, especially the clustered ones, were very likely to be initialized as fully potentiated, while synapses within the late assemblies were likely to be initialized in the fully depressed state (*Figure 6D* black arrows bottom). However, when comparing not the likelihood, but the amplitude of changes across categories with a two-way ANOVA, we found that clustered, within-assembly synapses depress to a significantly smaller degree than all other categories (*Figure 6—figure supplement 1A*). In contrast, non-clustered, within-assembly synapses (the ones that are most likely to change) depress the most. On the other hand, non-assembly synapses usually tended to be initialized depressed and to stay that way. Thus, the picture emerging is as follows: early and middle assemblies are partially defined by clustered (both spatial and functional) synapses that are initialized as fully potentiated. These synapses are unlikely

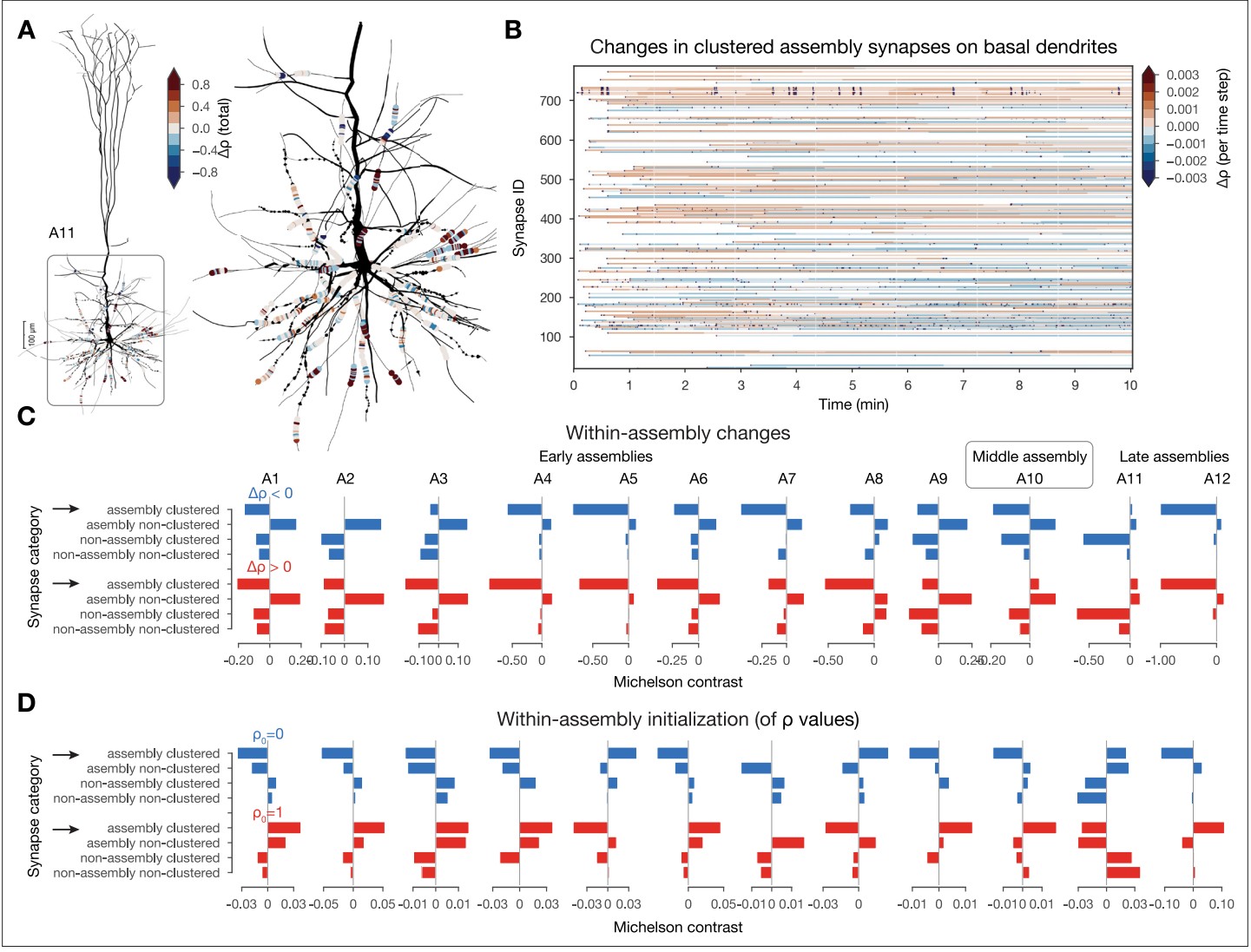

**Figure 6.** Interactions between plastic changes and synaptic clusters. (**A**) Changes in clustered assembly synapse on an exemplary neuron from assembly 11 (**A11**). Neurite diameters are scaled (2x) for better resolution. (Synapse diameters are arbitrary.) (**B**) Temporal evolution of the synapses on basal dendrites shown on A. Synapses are grouped by their dendritic sections (single stretches of non-branching dendrites) and ordered by their mean distance to the soma (closest on the bottom and furthest on top). (**C**) Michelson contrast (*Equation 15* in Methods) of the likelihood of plastic changes of synapses grouped into four categories: (non)-assembly and (non)-clustered for each assembly. Negative values indicate that synapses belonging to the given category are unlikely to change. Depression on top (in blue) and potentiation below (in red). Black arrows indicate clustered assembly synapses. (**D**) Same as C but for inital $\rho$ values (see Methods).

The online version of this article includes the following figure supplement(s) for figure 6:

**Figure supplement 1.** Changes in cross-assembly synapse clusters.

to change, but when they do, they depress less than the others, and would converge back to $\rho = 1$ in absence of activity, as they do not cross the $\rho_* = 0.5$ unstable fixed-point. These stable early assemblies can therefore function as a stable backbone amid ongoing synaptic plasticity.

In our previous investigation, we found that most changes happened across assemblies (*Figure 5F*), so we extended the analysis described above to cross-assembly synapses (*Figure 6—figure supplement 1B*). Here, the picture was reversed: cross-assembly synapses that were part of a spatial cluster were likely to be initiated as fully depressed and then had a high chance of undergoing potentiation (*Figure 6—figure supplement 1C* black arrow). Together with the previous results, this suggests that synapses between assemblies are more likely to change, which is even more pronounced if these synapses form a cluster on the postsynaptic dendrite.

# Network-level metrics predict plasticity and separate potentiation from depression

So far, we have found co-firing and synapse clustering to be good predictors of plastic changes, in line with previous work (*Zenke et al., 2015*; *Kastellakis and Poirazi, 2019*; *Harris, 2005*). Additionally, we have shown that changes are also driven by the underlying network's connectivity (*Figure 4A and B*). We now unfold the effect of network connectivity on plastic changes further, by studying the effect of the presynaptic population of the source (presynaptic) and target (postsynaptic) neurons of a connection on the $\rho$ dynamics. Note first that the indegree of a neuron, i.e., the size of its source population, is weakly correlated to its average firing rate (*Figure 7A*; $r = 0.13$). Furthermore, the indegree of the pre- and more strongly of the postsynaptic neurons of a connection are predictive of plastic changes (*Figure 7B1*). However, just like for assemblies, a purely rate-based control highlighted that the picture is more complex, firing rate alone is not the best predictor of plastic changes (*Figure 5—figure supplement 1B*).

Moreover, the correlation of the pre- and postsynaptic indegree ($r = 0.18$) complicates the (*Figure 7B2*). Thus, we took the natural choice of considering common innervation (see below), instead of making a choice of how to combine independent innervation of both neurons mediating the connection. Additionally, it has been shown that not only the size of the presynaptic population but also its internal connectivity affects co-firing and assembly formation (*Ecker et al., 2024*). We explore these effects using *k-edge indegree*, a novel yet simple network metric of edge centrality that considers how the edges (connections) are embedded in the entire network. More precisely, the $k$-edge indegree of a connection is the number of $k$-simplices that the connection is innervated by, where we call $k$ the dimension (*Figure 7C1*, see Methods). In particular, for $k = 0$ it is the number of neurons innervating both the pre- and post-synaptic neurons of the connection, i.e., a common neighbor as in *Perin et al., 2011*. Indeed, for our circuit, in the lower dimensions, $k$-edge indegree is correlated to the indegree of its source and target populations (*Figure 7—figure supplement 1A*; $r = 0.48, 0.35$ for $k = 0$ and $r = 0.35, 0.27$ for $k = 1$). We expected these motifs to have an effect on plasticity for two main reasons. First, pairs of connected neurons have higher spike correlations if the connection has higher $k$-edge indegree (*Reimann et al., 2017*), and the strength of this correlation depends on the overall higher-order structure of the network (*Nolte et al., 2020*). Second, simplex motifs innervating a cell have been shown to be a good predictor of assembly membership (*Ecker et al., 2024*), which in turn have an effect on plasticity as demonstrated above.

Across dimensions, as k-edge indegree increased so did the probability of undergoing plastic change (*Figure 7C2*). Furthermore, among the connections that change, the ones with the highest k-edge indegree tend to get potentiated while those with the lowest favor depression (*Figure 7C3*). Thus, we hypothesized k-edge indegree to be a good predictor of synaptic strength. We verified this both in our model as well as in the *MICrONS, 2021* mm³ dataset, an electron microscopic reconstruction of cortical tissue that combines synaptic resolution with the scale needed to calculate meaningful k-edge indegree. Both networks had similar maximal ddimensionsof simplices as well as ranges of k-edge indegree values (*Figure 7—figure supplement 1B*), making the results of this analysis comparable. For our model ,we used the sum of $\hat{g}_{AMPA}$ values as a proxy for connection strength. For MICrONS, we used total synapse volume of a connection (see Methods), as a measure of connection strength (*Harris and Stevens, 1989*). Indeed, high k-edge indegree led to higher connection strength in *MICrONS, 2021* as predicted by our modeling insights; moreover, the curves across dimensions are qualitatively the same for both datasets (*Figure 7D*).

To gain insights into pattern-specificity, we analyzed the 2-min-long single pattern simulations and considered k-edge indegree within the assembly-specific subgraphs (*Figure 7E1-Sub*). When doing so, we found an even higher predictive power, which was additionally pattern-specific (*Figure 7E2*). More precisely, the probability of changes only increased with k-edge indegree in the assembly subgraph associated with the presented pattern or to a late assembly and that it was the strongest for the former (see *Figure 7E2* left for pattern A for an example, right for a summary of the maxima attained for all patterns, and *Figure 4—figure supplement 1B* for consistency across repetitions of the same simulation).

Finally, *pattern-indegree*, i.e., the number of VPM fibers belonging to a pattern that innervate a neuron, is also a good predictor of the probability of a connection to change i.e., an increase of either the pre- or postsynaptic pattern-indegree of a connection leads to more frequent plastic changes

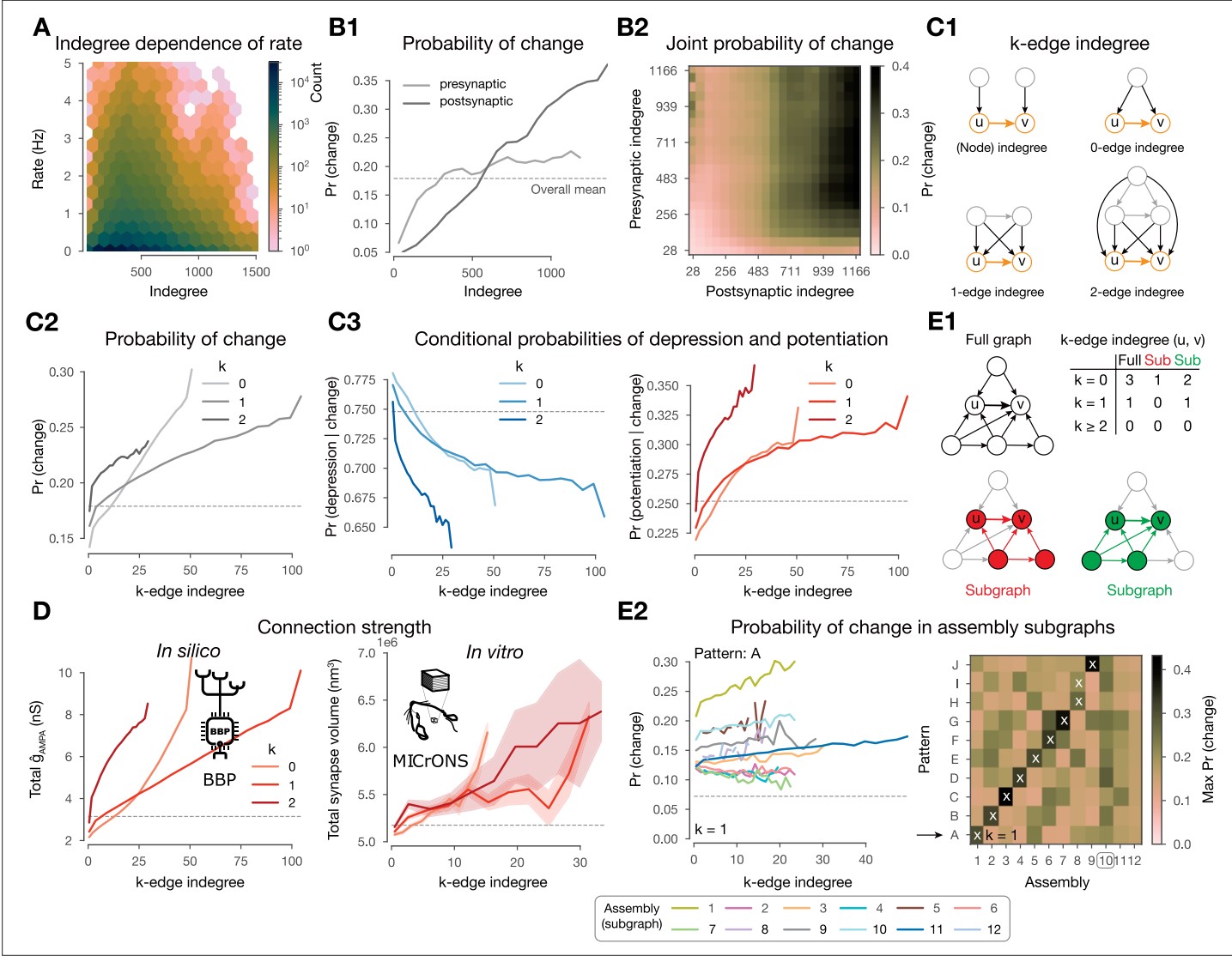

**Figure 7.** Changes are stronger in central connections in the network. (**A**) Firing rate vs. indegree of all excitatory neurons ($r = 0.13$). (**B**) Probability of plastic change vs. indegree. (**B1**) Separately for indegree of the pre- and postsynaptic neurons. (**B2**) Joint probability of change against the indegrees of both partners of the connection. (**C1**) Schematics of k-edge indegree across dimensions (k=0, 1, 2) and its difference from the classical (node) indegree. TODO: add something about colors. (**C2**) Probability of changes (i.e. either depression or potentiation) vs. k-edge indegree in high-dimensional simplices (see **C** and Methods). **C3** Probability of depression (left) and potentiation (right) conditioned on plastic change (in any direction) vs. k-edge indegree. (**D**) Comparison with electron microscopy data. Total $\alpha$-amino-3-hydroxy-5-methyl-4-isoxazolepropionate (AMPA) conductance ($\hat{g}_{AMPA}$) against k-edge indegree across dimensions in our nbS1 model on the left. Total synapse volume (see Methods) against k-edge indegree across dimensions in the (***MICrONS, 2021***) dataset on the right. Shaded areas indicate SEM. (**E1**) Schematics of k-edge indegree in high-dimensional simplices in an exemplary '*full*' network on the top in black and in (assembly) '*sub*'-graphs on the bottom. Table summarizes k-edge indegree values for the connection from node **u** to **v** across dimensions for all three cases. (**E2**) Probability of changes against k-edge indegree in assembly subgraphs (see C2 bottom). Left, probability of changes in (2-min-long) pattern A simulations against k-edge indegree in assembly subgraphs. Right, summary of the maximum probability values across patterns on the right. Arrow indicates the row shown in detail on its left and white crosses indicate the pattern-specific early assemblies (see ***Figure 5B***).

The online version of this article includes the following figure supplement(s) for figure 7:

**Figure supplement 1.** Correlation of indegree and k-edge indegree, structural comparison with MICrONS and pattern-indegree analysis.

(***Figure 7—figure supplement 1C1***1). However, given the highly non-random structure of the full network, the values of pre- and postsynaptic pattern-indegree are not independent and thus their effect in driving plastic changes is hard to disentangle (***Figure 7—figure supplement 1C2 and D***; $r = 0.230$. on average).

In summary, k-edge indegree in the full network allowed us to distinguish potentiation from depression given change, as well as predict connection strength, which we confirmed in the (*MICrONS, 2021*) dataset. Moreover, k-edge indegree within assembly subgraphs is a pattern-specific predictor of plastic changes.

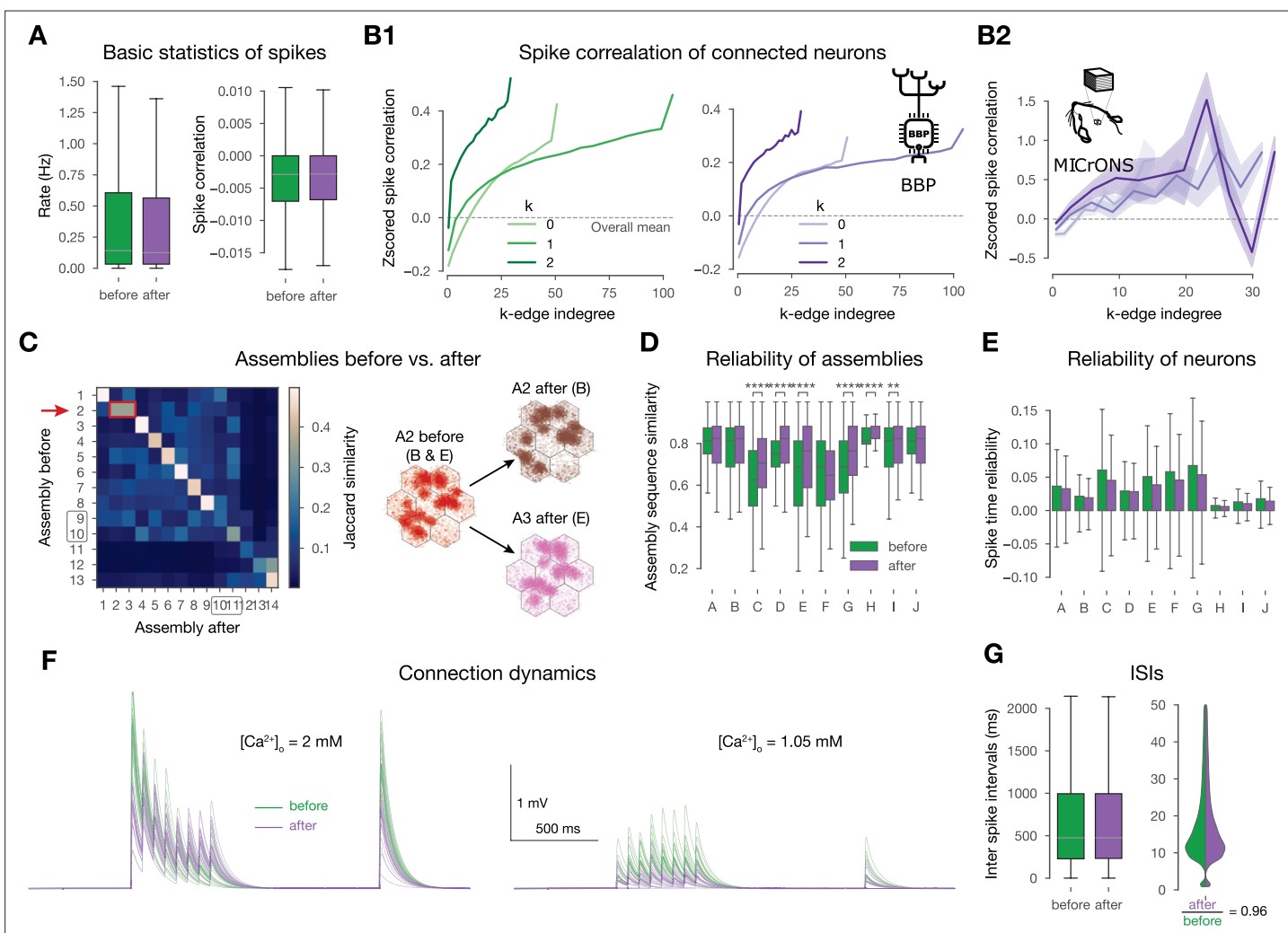

**Figure 8.** Changes promote stimulus specificity. (**A**) Firing rates and pairwise spike correlations extracted from non-plastic simulations *before* plasticity, i.e., in the naive circuit vs. *after* the 10-min-long plastic simulation. (**B1**) Spike correlation of connected cells vs. k-edge indegree in high-dimensional simplices (see Methods) before and after plasticity on the left and right, respectively. (**B2**) Spike correlation of connected cells vs. k-edge indegree in the (*MICrONS, 2021*) dataset (see Methods). Shaded areas indicate SEM. (**C**) Jaccard similarity of assemblies detected before vs. after plasticity on the left. Shared early assembly of patterns B and E splitting into two stimulus-specific ones after plasticity on the right. (Indicated by red arrow and rectangle on its left). See more detailed plots in figure-supplement 1B. (**D**) Reliability of assembly sequences (see Methods and figure-supplement 1B). Significance of increases was determined with the Kruskal-Wallis test: *: $p \leq 0.05$, **: $p \leq 0.01$, ***: $p \leq 0.001$, ****: $p \leq 0.0001$. (**E**) Spike time reliability (see Methods) of single cells to the different patterns before and after plasticity. (The lack of stars means no significant increases after plasticity.) (**F**) Short-term dynamics of an exemplary potentiated L5 TTPC connection before (in green) and after plasticity (in purple). At in vitro $[Ca^{2+}]_o$ on the left, and in vivo on the right. Thin lines represent 20 individual trials, while the thicker ones represent their means. (**G**) Interspike interval (ISI) distribution of all excitatory neurons before and after plasticity on the left. Zoom in on low ISIs ($\leq 50$ ms) on the right.

The online version of this article includes the following figure supplement(s) for figure 8:

**Figure supplement 1.** Layer-wise changes and assemblies detected before and after plasticity.

**Figure supplement 2.** Main results with slightly different plasticity model parameters.

## Increased stimulus-specificity and more reliable assembly sequences characterize the network after plasticity

As the changes are sparse, one might wonder if they cause any functional change. To study this, we compared the network's activity *before* and *after* plasticity (see Methods). Firing rates decreased slightly as a result of plasticity (*Figure 8A* left), while spike correlations remained stable, in line with recent findings (*Oby et al., 2019*; *Feulner et al., 2022*, *Figure 8A* right). Both of these effects were more pronounced in the deeper layers (*Figure 8—figure supplement 1A*). Correlations of connected pairs of neurons decreased slightly after plasticity following the decrease in firing rate (*Figure 8—figure supplement 1A*). However, the highest correlations both before and after plasticity were found in the most central edges in the network i.e, those with highest edge participation in high-dimensional simplices (*Figure 8B1*). We further verified this prediction in the MICrONS dataset (*Figure 8B2*, see Methods). It has been previously shown that assemblies have many edges that are central in the network, therefore, we hypothesized that they remain functional and reliably active (*Fauth and van Rossum, 2019*; *Kossio et al., 2021*; *Pérez-Ortega et al., 2021*). We showed this on the level of assembly sequences: plastic changes promote stimulus-specific and reliable responses. While we found considerable overlap between assemblies detected before and after plasticity, an early assembly responding to patterns B and E before plasticity, split into two stimulus-specific assemblies (*Figure 8C*). Note that this is not an artifact of the thresholding applied during assembly detection as forcing the pipeline to yield the same number of assemblies instead splits a middle assembly first (*Figure 8—figure supplement 1E*). Moreover, the assembly sequences responding to repetitions of a given pattern generally became more reliable after plasticity (*Figure 8—figure supplement 1B*). To quantify this, we measured the Hamming similarity of assembly sequences between repetitions and found significant increases for most patterns (see Methods, *Figure 8D*). Interestingly, this increase on the assembly level lied in contrast with a slight decrease in the reliability of the neuronal activity on the single cell level (*Figure 8E*).

The plastic changes we observed at the network level are in contrast with in vitro observations at the connection level. In particular, with the redistribution of synaptic efficacy towards earlier spikes during high-frequency firing caused by changes in $U_{SE}$ (*Chindemi et al., 2022*; *Markram and Tsodyks, 1996*; *Costa et al., 2015*; *Selig et al., 1999*; *Sjöström et al., 2003*), which under our in vivo-like, low firing rates and $[Ca^{2+}]_o$ is not in the ranges where this connection-level redistribution is relevant. Specifically, at in vitro levels of $[Ca^{2+}]_o$ the potentiated $U_{SE}$ shifts the connection further into the depressing regime, causing the redistribution (*Figure 8F* left). In vivo the lower $[Ca^{2+}]_o$ counters this by moving the connection into the pseudo-linear regime; additionally, the low firing rates make this phenomenon less relevant in general (*Figure 8F* right and G, respectively). Thus, while *Markram and Tsodyks, 1996* showed a redistribution of synaptic efficacy after plasticity at the single connection level in vitro (*Figure 8F*), we found a redistribution at the network level under in vivo-like conditions (*Figure 8A-D*, *Figure 8—figure supplement 1A*).

In summary, we observed a network-level redistribution of synaptic efficacy that maintained spike correlations globally, while enhancing stimulus-specificity and the reliability of the subplopulations in terms of assembly activation. Furthermore, we derived and confirmed in the (*MICrONS, 2021*) dataset a prediction stating that the most central connections of the network drive the correlated activity.

## Discussion

In this work, we aimed to understand how network structure, function, and dendritic processing interact to drive synaptic plasticity at the population level in the neocortex. Specifically, we used a plasticity model, whose parameters were determined by pairwise STDP experiments but modeled at the level of dendrites. We studied its effect at the population level under in vivo-like conditions to determine network-level rules that best predict plastic changes by simulating it in a biophysically detailed model of a cortical microcircuit that was subjected to 10 different stimuli under a variety of protocols. Our principal observations are as follows: (1) Plastic changes were sparse, affecting only 7% of the synapses. A balance between occasional, large-amplitude potentiation and more frequent depression stabilized the network without explicitly modeling homeostatic plasticity. Moreover, the changes were non-random and stimulus-specific. (2) Plastic changes were largely determined by the

anatomical structure of synaptic connectivity and its relation to functional units, i.e., changes were most likely between co-firing cell assemblies, at clustered synapses, between neurons highly innervated by the same input, and in central edges in the network. (3) While assemblies remained fairly stable, they became more stimulus-specific and reliable which has important implications for coding and learning.

The first observation (1) is quite significant considering that we did not design the learning rule to be sparse and stable. In previous models of plastic networks, changes were not sparse and additional mechanisms were needed to keep the network's activity stable (*Litwin-Kumar and Doiron, 2014*; *Delattre et al., 2015*; *Zenke et al., 2015*; *Fauth and van Rossum, 2019*; *Kossio et al., 2021*; *Zenke et al., 2017a*; *Turrigiano and Nelson, 2004*). The machine learning community is also aware of the importance of sparse changes, as in continual learning one has to balance plasticity and stability of existing weights to avoid catastrophic forgetting (*McCloskey and Cohen, 1989*; *Ratcliff, 1990*). In recent years, they have come up with impressive techniques that mask or slow down weight updates (e.g. by a regularizer on the loss) to improve the performance of deep neural networks (*Zenke et al., 2017b*; *Kirkpatrick et al., 2017*; *Mallya and Lazebnik, 2018*; *Frankle and Carbin, 2019*), whereas in our model it emerged without any mechanisms explicitly enforcing it. Note, however, that despite the comparison to machine learning, our model is not designed to solve any specific task. Instead, it is a model of sensory cortex which only learns to represent stimulus features in an unsupervised manner. Nonetheless, demonstrating the stimulus-specificity of changes as well as an increase in the reliability of assembly sequences, was a crucial validation that our results are not just a description of a biophysical process, i.e., plasticity, but have implications for learning in general.

For our second observation (2), we described three different kinds of plasticity rules that emerge from the underlying plasticity model based on: activity, subcellular structure, and network structure. While all of these are powerful predictors of plastic changes, none of them fully determines them. It is rather the interplay between them (and potentially additional unknown rules) that brings about non-homogeneous changes across the full microcircuit. The first two can be explained from the biophysics of the plasticity model and links our results to the classical work of *Hebb, 1949* as well as the recent literature on synapse clustering (*Kastellakis and Poirazi, 2019*; *Farinella et al., 2014*; *Iacaruso et al., 2017*; *Tazerart et al., 2020*). With respect to synapse clustering, we would highlight that our synapses are stochastic and the release probability between PCs is ~ 0.04 at the simulated low $[Ca^{2+}]_o = 1.05$ mM (*Markram et al., 2015*; *Ecker et al., 2020*; *Jones and Keep, 1988*; *Borst, 2010*). Therefore, care should be taken when comparing our results with either glutamate uncaging experiments, which bypass the presynaptic machinery (*Pettit et al., 1997*; *Losonczy and Magee, 2006*), or with other modeling studies that use deterministic synapses (*Farinella et al., 2014*; *Poirazi et al., 2003*; *Ujfalussy and Makara, 2020*). With respect to network-based rules, previous simulation approaches have characterized the learned connectomes as having an overexpression of reciprocal connectivity and densely connected neuron motifs (*Brunel, 2016*; *Zhang et al., 2019*). We expanded on this body of previous work, by introducing cell and connection-specific metrics that directly yield the probability of observing depression or potentiation of any given connection. Most of our predictors behaved similarly with respect to depression and potentiation. Only a metric based on the embedding of individual edges in the global (or assembly-specific) network was able to separate them. This edge-based prediction was confirmed by analyzing an electron microscopic reconstruction of cortical tissue in a comparable way (*MICrONS, 2021*).

Our third observation (3) is that assembly sequence becomes more reliable and stimulus-specific may sound counter-intuitive considering that we observed reduced firing rates and less reliable single neurons. This shows that the plasticity mechanism can contribute to the emergence of a robust population code with unreliable neurons. Crucially, this means that plasticity can make circuit responses more distinct (between stimuli by increasing specificity) but also less distinct (for a given stimulus by increasing reliability). This has important computational implications that may be explored in a future study.

To facilitate further research, we are open-sourcing our model alongside detailed instructions to launch simulations and analyze the results (see Data and code availability). Simulating the model requires a performant hardware and software infrastructure. With respect to the latter, we are continuously improving the performance and efficiency of the simulator (*Kumbhar et al., 2019*). The model has several assumptions and limitations, which will ideally be improved upon iteratively and in a

community-driven manner. First, by using the circuit model of *Isbister et al., 2023*, we inherit all assumptions listed therein; one of the most important one being on source data. A common limitation of data-driven modeling is the lack of high-quality source data, thus while we refer to our model as juvenile rat non-barrel somatosensory cortex, sometimes we used data from the barrel cortex, from adult animals and from mouse. Second, for our simulations we used the model of *Chindemi et al., 2022* with parameters obtained in a previous version of the circuit. Due to various improvements of synapse parameters (e.g. $U_{SE}$ of L5 PCs) since fitting plasticity-specific parameters, the results for pairing protocols at various frequencies shift slightly while maintaining the presence of STDP (*Figure 1—figure supplement 2A*). When exploring slightly different parameters of the plasticity model, we found that while the exact numerical values changed, our main conclusions about nonrandom changes and various features predicting them were not affected (*Figure 8—figure supplement 2*.) Third, the extracellular magnesium concentration of 1 mM used in in vitro preparations was assumed to be representative of the in vivo level. Fourth, experimental evidence indicates that firing bursts affects plasticity (*Letzkus et al., 2006*; *Williams and Stuart, 1999*; *Larkum, 2013*). However, L5 TTPC bursts were rare in our simulations, as the model is based on an early developmental stage (P14-16: juvenile rats) and firing bursts only becomes prominent as the animals mature (*Zhu, 2000*). Furthermore, bursts can be triggered by synaptic input onto apical dendrites (*Larkum, 2013*; *Hay et al., 2011*). Inputs to apical dendrites from high-order thalamus have been shown to gate plasticity in L23 PCs in vivo, via disinhibiting the distal dendrites (*Gambino et al., 2014*; *Williams and Holtmaat, 2019*). While POm input was present in our model and targeted apical dendrites, we activated them in random and non-specific ways. On the other hand, top-down inputs represent context or brain state and are thought to serve as an error or target signal for learning and thus are likely very specific (*Makino, 2019*). Feedback signals from high-order cortical areas also innervate apical dendrites (*Harris et al., 2019*), but as our model is of a single cortical area only, these inputs were absent. Missing excitatory input from other areas were compensated by somatic conductance injections (*Isbister et al., 2023*), which could be replaced in the future by simulating additional dendritic synapses instead. Finally, learning in the brain is orchestrated by several mechanisms in unison and some of these are beyond the scope of this paper, e.g., inhibitory plasticity, metabotropic glutamate receptors, neuromodulation, slow homeostatic plasticity, structural plasticity, and calcium-induced calcium release (*Magee and Grienberger, 2020*).

Bottom-up modeling is often criticized for being arbitrarily complex. In our simulations, the mean value of synaptic efficacies of an over-excited network reduced, comparable to *Turrigiano and Nelson, 2004* without introducing rate homeostasis explicitly. In the machine learning field, mechanisms are added to classical algorithms to make updates sparser, overcome catastrophic forgetting, and allow the learning of tasks sequentially (*Kirkpatrick et al., 2017*; *Zenke et al., 2017b*). The calcium-based plasticity rule employed here results in sparse updates without additional mechanisms, unlike STDP rules, which update efficacies at each spike arrival. Thus, we showed that adding biological complexity allowed us to reduce the number of mechanisms explicitly modeled. Calcium-based plasticity models existed for a long time (*Shouval et al., 2002*; *Rubin et al., 2005*), have been shown to be stable (*Higgins et al., 2014*; *Graupner et al., 2016*; *Wang and Aljadeff, 2022*), have been modified to mimic in vivo-like low $[Ca^{2+}]_o$ (*Graupner et al., 2016*; *Chindemi et al., 2022*), and versions more complex than ours exist (*Mäki-Marttunen et al., 2020*; *Rodrigues et al., 2023*). Our contribution lies in integrating all these different aspects in network simulations at scale with subcellular resolution. Using such simulations allowed us to explore not only the temporal aspects of plasticity but also their link to spatial and network aspects, such as clusters of synapses and cell assemblies. Future work could explore the constellation of changing connections when stimuli change or cease, providing a more concrete examination of continuous learning paradigms within this model. Furthermore, the analysis methods developed on large-scale functional and structural data empowered us to make and test novel predictions in the (*MICrONS, 2021*) dataset, which while pushing the boundaries of big data neuroscience, was so far only analyzed with focus on either single cells or first-order network metrics (e.g. node degree) (*Ding et al., 2024*; *Wang et al., 2023*). We extend this work by considering the higher-order structure of the network.

## Methods

**Key resources table**

| Reagent type (species) or resource | Designation | Source or reference | Identifiers | Additional information |
|---|---|---|---|---|
| Software, algorithm | Plastyfire | this paper | https://github.com/BlueBrain/plastyfire | Calcium-based plasticity model |
| Software, algorithm | Neurodamus | *Isbister et al., 2023* | https://github.com/BlueBrain/neurodamus | |
| Software, algorithm | AssemblyFire | *Ecker et al., 2024* | https://github.com/BlueBrain/assemblyfire | |
| Software, algorithm | Connectome Utilities | *Reimann et al., 2024b* | https://github.com/BlueBrain/Connectome-Utilities | |
| Software, algorithm | Connectome Analysis | *Santander et al., 2024* | https://github.com/BlueBrain/connectome-analysis | |
| Software, algorithm | Zenodo Dataset | this paper | https://doi.org/10.5281/zenodo.8158471 | Data and code availability |

### Resource availability

#### Lead contact
Further information and requests for data and code should be directed to and will be fulfilled by the lead contact: Michael W. Reimann (mwr@reimann.science).

#### Materials availability
No materials were used in this computational work.

### Method details

#### Calcium-based plasticity model
The calcium-based long-term plasticity model used for E to E connections is fully described in *Chindemi et al., 2022*, but a minimal description of it can be found below. Synaptic efficacy ($\rho$) is based on the (*Graupner and Brunel, 2012*) formalism, which exhibits a bistable dynamics ($\rho = 0$ fully depressed, $\rho = 1$ fully potentiated, and $\rho_* = 0.5$ unstable fixed-point) described as:

$$\tau \frac{d\rho}{dt} = -\rho(1-\rho)(\rho_* - \rho) + \gamma_p(1-\rho)\Theta\left(Ca^* - \theta_p\right) - \gamma_d\rho\Theta\left(Ca^* - \theta_d\right) \tag{1}$$

where $\tau = 70$ s is the time constant of convergence, $\theta_d$ and $\theta_p$ are depression and potentiation thresholds, $\gamma_d = 101.5$ and $\gamma_p = 216.2$ are depression and potentiation rates and $\Theta$ is the Heaviside function. $Ca^*$ is linked to the dynamics of $[Ca^{2+}]_i$ in spines (see below), which was modeled as:

$$\frac{d[Ca^{2+}]_i}{dt} = \left(I^*_{NMDA} + I_{VDCC}\right)\frac{\eta}{2FX} - \frac{[Ca^{2+}]_i - [Ca^{2+}]_i^{(0)}}{\tau_{Ca}} \tag{2}$$

where $I^*_{NMDAR}$ and $I_{VDCC}$ are calcium currents through NMDA receptors and VDCCs, $\eta = 0.04$ is the fraction of unbuffered calcium, $F$ is the Faraday constant, $X$ is the spine volume, $[Ca^{2+}]_i^{(0)} = 70$ pM is the resting value of $[Ca^{2+}]_i$, and $\tau_{Ca} = 12$ ms is the time constant of free (unbuffered) calcium clearance. $I^*_{NMDA}$ depends on the state of the Mg$^{2+}$ block as follows:

$$I_{NMDA}(t) = g_{NMDA}m(V - E_{NMDA}) \tag{3}$$

where $g_{NMDA}$ and $E_{NMDA} = -3$ mV are the conductance and the reversal potential of the NMDA receptor, $V$ is the local voltage, and $m$ describes the nonlinear voltage dependence due to the Mg$^{2+}$ block following the (*Jahr and Stevens, 1990*) formalism:

$$m = \frac{1}{1 + [Mg^{2+}]_o/\theta e^{-\kappa V}} \tag{4}$$

where $\theta$ is a scaling factor of the extracellular magnesium concentration $[Mg^{2+}]_o$, and $\kappa$ is the slope of the voltage dependence. Parameters $\theta = 2.552$ and $\kappa = 0.072$ were obtained by refitting the model to

cortical recordings from *Vargas-Caballero and Robinson, 2003* (as opposed to the original parameters fit to hippocampal ones *Jahr and Stevens, 1990*). Spines were assumed to be separate biochemical compartments, i.e., $[Ca^{2+}]_i$ of the dendrites does not influence that of the synapses.

Inspired by previous theoretical insights (*Rubin et al., 2005*), a leaky integrator of $[Ca^{2+}]_i$ was introduced ($Ca^*$) to slow down its time course instead of modeling enzymes downstream of calcium (e.g. CamKII as others did *Mäki-Marttunen et al., 2020*; *Rodrigues et al., 2023*):

$$\frac{dCa^*}{dt} = -\frac{Ca^*}{\tau^*} + [Ca^{2+}]_i - [Ca^{2+}]_i^{(0)} \tag{5}$$

where $\tau^* = 278.318$ ms is the time constant of the integrator. Updates in $\rho$ were linked to this $Ca^*$ variable crossing $\theta_d$ and/or $\theta_p$ (see *Equation 1*). The two synapse-specific thresholds were derived based on peaks in $[Ca^{2+}]_i$ caused by pre- and postsynaptic spikes, $c_{pre}$ and $c_{post}$, respectively. To measure these parameters for all E to E 312,709,576 synapses, simulations of single cells were run, in which either the pre- or the postsynaptic excitatory cell was made to fire a single action potential and the local $[Ca^{2+}]_i$ was monitored in each synapse. Presynaptically evoked $[Ca^{2+}]_i$ peaks were three orders of magnitude larger, than the ones evoked by postsynaptic spikes (*Figure 1—figure supplement 2B*). Postsynaptically evoked $[Ca^{2+}]_i$ peaks had a multimodal distribution in the apical dendrites (*Figure 1—figure supplement 2B* right), in line with (*Landau et al., 2022*). Since 8% of L6 PCs could not be made to fire a single action potential (only bursts), synapses on those cells (10,995,513 in total) were assumed to be non-plastic, i.e., their thresholds were set to a negative value that could not be crossed. Similarly, as the plasticity of connections between L4 spiny stellate cells was shown to be non-NMDA dependent (*Chindemi et al., 2022*; *Egger et al., 1999*) those connections were made non-plastic. For the remainder of cells, $\theta_d$ and $\theta_p$ were derived as follows:

$$\begin{bmatrix} \theta_d \\ \theta_p \end{bmatrix} = \begin{bmatrix} a_{00} & a_{01} \\ a_{10} & a_{11} \end{bmatrix} \times \begin{bmatrix} c_{pre} \\ c_{post} \end{bmatrix} \text{ or } \begin{bmatrix} b_{00} & b_{01} \\ b_{10} & b_{11} \end{bmatrix} \times \begin{bmatrix} c_{pre} \\ c_{post} \end{bmatrix} \tag{6}$$

where $a_{i,j}$ and $b_{i,j}$ are constants optimized during model fitting for apical and basal dendrites, respectively. Changes in $\rho$ were then converted by low-pass filtering into changes $U_{SE}$ and $\hat{g}_{AMPA}$ as follows:

$$\frac{dU_{SE}}{dt} = \frac{\overline{U}_{SE} - U_{SE}}{\tau_{change}} \quad \text{where} \quad \overline{U}_{SE} = U_{SE}^{(d)} + \rho\left(U_{SE}^{(p)} - U_{SE}^{(d)}\right) \tag{7}$$

$$\frac{d\hat{g}_{AMPA}}{dt} = \frac{\overline{g}_{AMPA} - \hat{g}_{AMPA}}{\tau_{change}} \quad \text{where} \quad \overline{g}_{AMPA} = \hat{g}_{AMPA}^{(d)} + \rho\left(\hat{g}_{AMPA}^{(p)} - \hat{g}_{AMPA}^{(d)}\right) \tag{8}$$

where $U_{SE}^{(d)}$, $U_{SE}^{(p)}$, $\hat{g}_{AMPA}^{(d)}$, and $\hat{g}_{AMPA}^{(p)}$ are the fully depressed (d) and fully potentiated (p) values of the given variables in-between which they evolve and $\tau_{change} = 100$ s is the time constant of the conversion of changes in $\rho$ into changes in $U_{SE}$ and $\hat{g}_{AMPA}$.

In the nbS1 model $U_{SE}$ is also modulated by $[Ca^{2+}]_o$, where a reduction in $[Ca^{2+}]_o$ leads to a pathway-specific, non-linear reduction in $U_{SE}$ (*Markram et al., 2015*; *Ecker et al., 2020*).

When extracting the network's state after plasticity, not only the $U_{SE}$ and $\hat{g}_{AMPA}$ values, but also the peak NMDA conductances ($\hat{g}_{NMDA}$) were updated according to the $\rho$ values in the last time step of the simulation.

Model parameter optimization (*Chindemi et al., 2022*) and the derivation of thresholds from $c_{pre}$ and $c_{post}$ measurements was done with plastyfire.

## In vivo-like spontaneous and evoked activity

The calibration process that leads to the in vivo-like spontaneous activity is fully described in *Isbister et al., 2023*, but a minimal description and a list of the parameters used in this article can be found below. As extracellular recordings are known to overestimate firing rates (*Wohrer et al., 2013*), a spectrum of spontaneous states at fixed percentage of the rates reported in *Reyes-Puerta et al., 2015* were calibrated (*Isbister et al., 2023*). Matching specific firing rates in silico was achieved by iterative adjustments of layer and cell-type (excitatory/inhibitory) specific somatic conductance injection (following an Ornstein-Uhlenbeck process *Destexhe et al., 2001*). As in *Isbister et al., 2023*, extracellular recordings were assumed to have the same bias across layers and neuron populations.

Furthermore, it was assumed that different inhibitory subpopulations require the same amount of input compensation. By introducing plasticity at all E to E synapses, an additional depolarizing current from VDCCs was added to the model, which made the network more active than its non-plastic counterpart (*Figure 1—figure supplement 3A*). This required an algorithmic lowering of the amplitude of injected conductances from *Isbister et al., 2023* to achieve the same in vivo-like layer-wise spontaneous firing rates (*Figure 1—figure supplement 3B*). The spontaneous state used in the article is characterized by the parameters: $[Ca^{2+}]_o$ = 1.05 mM (*Jones and Keep, 1988*), percentage of reported firing rates = 40%, the coefficient of variation (CV; std/mean) of the noise process = 0.4. As in *Markram et al., 2015*, synapse dynamics (also known as short-term plasticity) of all synaptic pathways (not only E to E) played a pivotal role in achieving an in vivo-like asynchronous firing regime. In short, the modulation of the baseline release probability ($U_{SE}$) by $[Ca^{2+}]_o$ is pathway-dependent in the model, affecting E to E synapses more than I to E ones. This way, the I to E synapses have a smaller reduction in their efficacy at low $[Ca^{2+}]_o$ and, therefore, put the network in an E/I balanced state (*Markram et al., 2015*).

The thalamic input patterns, and the spike trains delivered on them are fully described in *Ecker et al., 2024*, but a minimal description, highlighting the changes applied in this study, can be found below. First, the flatmap location (*Bolaños-Puchet et al., 2024*) of VPM fibers avoiding the boundaries of the network were clustered with k-means to form 100 bundles of fibers. Second, the four base patterns (A, B, C, and D) were formed by randomly selecting four non-overlapping groups of bundles, each containing 12% of them. The remaining six patterns were derived from these base patterns with various degrees of overlap: three patterns as combinations of two of the base ones (E, F, G), two patterns as combinations of three of the base ones (H, I), and one pattern as a combination of all four base ones (J). Third, the input stream was defined as a random presentation of these 10 patterns, in a balanced way. Last, for each pattern presentation, unique spike times were generated for its corresponding fibers following a 100 ms-long inhomogeneous adapting Markov process (*Muller et al., 2007*). The maximal rate of the VPM fibers was set to 17.5 Hz (compared to 30 Hz for the non-plastic circuit in *Ecker et al., 2024*) and half of that for POm. The overlap of the patterns is clearly visible in the firing pattern of each group of fibers corresponding to them (*Figure 1—figure supplement 4*).

## Network simulations

Simulations were run using the NEURON simulator as a core engine with the Blue Brain Project's collection of hoc and NMODL templates for parallel execution on supercomputers (*Kumbhar et al., 2019*; *Hines and Carnevale, 1997*; *Awile et al., 2022*) (see Data and code availability). Simulating 10 min of biological time with reporting the state of all synapses (in every second) took 2,350,000 core hours ($\sim 4x$ more than the corresponding non-plastic circuit without reporting), on our HPE-based supercomputer, installed at CSCS, Lugano. Although deep profiling of all campaigns was not done, an estimated 12 M core hours were required to run all simulations presented in the manuscript. Simulations were always repeated at least three times to assess the qualitative consistency of the results (*Figure 4—figure supplement 1*).

## Control STDP rule

To compare the amount of changes induced by the calcium-based model of *Chindemi et al., 2022* with classical plasticity rules, the 35,264,818 excitatory spikes from the 10-min-long simulation were evaluated with a pair-based STDP rule (*Gerstner et al., 1996*; *Kempter et al., 1999*; *Song et al., 2000*). Synaptic weights evolved as follows under the STDP rule:

$$\Delta w_+ = A_+ \exp\left(-\frac{\Delta t}{\tau_+}\right) \text{ at } t_{post} \text{ if } t_{pre} < t_{post} \tag{9}$$

$$\Delta w_- = A_- \exp\left(\frac{\Delta t}{\tau_-}\right) \text{ at } t_{pre} \text{ if } t_{pre} > t_{post} \tag{10}$$

where $t_{pre}$ and $t_{post}$ are the times of pre- and postsynaptic spikes, $\Delta t = t_{post} - t_{pre}$ is the difference between them; $A_{\pm} = 0.05$ describe the weight update, which decayed exponentially with time constants $\tau_{\pm} = 20$ ms. The same parameters were used for the comparison with the 10-min-long synchronous (unstable) simulation (76,025,009 excitatory spikes). The STDP rule was implemented in Brian2 (*Stimberg et al., 2019*).

## Random walk control

Changes in $\rho$ induced by the calcium-based model of *Chindemi et al., 2022* and the STDP rule described above, were compared to a random control that takes into account global and noise-induced random changes in the network. To that end, a control model was used, where $K$ connections of the network either increase or decrease by a noise value ε with equal probability in each time step. Note that, in this control, the size of the change vector at each time step is given by:

$$\|\rho(t + \Delta t) - \rho(t)\|_2 = \epsilon \sqrt{K}. \tag{11}$$

This process was modeled with a *random walk* in $K$-dimensional space of step size $l$, which at step $N$ was given by:

$$\vec{R} = \sum_{i=0}^{N} \vec{r_i}, \tag{12}$$

where each $\vec{r_i}$ is a random variable in $\mathbb{R}^K$, where $K$ is the number of changing connections of the network and such that the step size is given by $|\vec{r_i}| = \epsilon\sqrt{K} =: l$. The changes induced by the random walk at step $N$ in the network is given by its distance from the origin at that time step, which for large enough $N$ and $K$ is approximately:

$$\|\vec{R}\| \approx l\sqrt{N}. \tag{13}$$

The step size $l$ was determined as the average step size in the data after the 5 min mark, when the transient activity has long passed. That is, $l$ is given by:

$$\underset{t>5\text{minutes}}{\text{mean}} \{\|\rho(t + \Delta t) - \rho(t)\|_2\}. \tag{14}$$

## Cell assembly detection

The combination of methods from *Carrillo-Reid et al., 2015* and *Herzog et al., 2021* yielding the assembly detection pipeline is fully described in *Ecker et al., 2024*, but a minimal description, highlighting the changes applied in this study, can be found below. First, spikes of excitatory cells were binned using 20 ms time bins (*Harris et al., 2003*). Second, time bins with significantly high firing rates were determined as crossing a threshold defined as the mean activity level plus the 95th percentile of the standard deviation of 100 shuffled controls. These shuffled controls were less strict than in *Ecker et al., 2024*. Unlike in the original study, where spikes were only shifted by one time bin forward or backward (*Carrillo-Reid et al., 2015*), spikes were shifted by any amount. This change was introduced because the network's response to the same patterns was more variable in the plastic simulations, and to not miss any of them, a lower threshold was more fitting. Third, based on the cosine similarity of activation vectors, i.e., vectors of spike counts of all neurons in the given significant time bins, a similarity matrix was built (*Carrillo-Reid et al., 2015*). Fourth, this similarity matrix was hierarchically clustered using Ward's linkage (*Pérez-Ortega et al., 2021*; *Montijn et al., 2016*). Like for any other unsupervised clustering method, the number of optimal clusters cannot be known beforehand, thus potential number of clusters were scanned between five and twenty. The number of clusters with the lowest Davis-Bouldin index was chosen, which maximizes the similarity within elements of the cluster while minimizing the between cluster similarity (*Davies and Bouldin, 1979*). Fifth, neurons were associated towiththese clusters based on their spiking activity, and it was determined whether they formed a cell assembly or not. The correlations between the spike trains of all neurons and the activation sequences of all clusters were computed and the ones with significant correlation selected to be part of the potential assemblies. Significance was determined based on exceeding the 95th percentile of correlations of shuffled controls (1000 controls with spikes of individual cells shifted by any amount as above; *Herzog et al., 2021*; *Montijn et al., 2016*). Finally, it was required that the mean pairwise correlation of the spikes of the neurons with significant correlations was higher than the mean pairwise correlation of neurons in the whole dataset (*Herzog et al., 2021*). Clusters passing this last criterion were considered to be functional assemblies and the neurons with significant correlations their constituent cells.

The reliability of assembly sequences was defined as the Hamming similarity over the repetitions of a single pattern, which attains a value of 1 if two assembly sequences are identical and 0 if they are completely different. Assemblies of neurons were compared using their Jaccard similarity.

Assemblies were detected using the assemblyfire package.

## Synapse clusters and likelihood of plastic changes within them

To quantify the importance of co-localization of synapses for plastic changes, clusters of synapses were detected. To be part of a synapse cluster, a synapse was required to have at least nine other synapses on the same dendritic branch, i.e., between two branching points of the dendrite, with $\leq 10\,\mu m$ (Euclidean) distance. Significance of spatial clustering was determined similarly to *Druckmann et al., 2014*. To that end, the distribution of synapse neighbor distances of the 10 selected synapses were compared with a Poisson model, assuming exponentially distributed inter-synapse distances, based on all (same branch) synapse neighbor distances on the given neuron. Clusters were merged in a subsequent step, thus synapse clusters with more than 10 synapses, spanning more than 20 μms were also feasible.

Plastic changes in synapse clusters were only analyzed for a small subpopulation of neurons in an assembly (10 L5 TTPCs per assembly), which were selected based on maximizing two connectivity features: assembly-indegree and synaptic clustering coefficient (SCC). The first one is the number of connections from a presynaptic assembly, while the second one quantifies the co-localization of synapses on the dendrites of a neuron from its presynaptic assembly. Maximizing these two metrics, which were introduced (and are described in detail) in *Ecker et al., 2024*, allows one to select subpopulations with high probability of finding synapse clusters.

Control synapse clusters, originating from non-assembly neurons were also detected on the same postsynaptic neurons.

The normalized likelihood of changes, conditioned on the four *categories* a synapse could fall into (assembly clustered, assembly non-clustered, non-assembly cluster, non-assembly non-clustered) were quantified using the Michaelson contrast, defined as:

$$\frac{Pr(changed\,|\,category) - Pr(changed)}{Pr(changed\,|\,category) + Pr(changed)} \tag{15}$$

where *changed* was split to be either potentiated ($\Delta\rho > 0$) or depressed ($\Delta\rho < 0$). Note that a nonzero value for one category always has to be compensated by a nonzero value with the opposite sign in another. For the normalized likelihood of initial $\rho$ values, the same equation was used but $Pr(\Delta\rho > 0\,|\,category)$ was replaced with $Pr(\rho_0 = 1\,|\,category)$ for potentiated and $Pr(\Delta\rho < 0\,|\,category)$ with $Pr(\rho_0 = 0\,|\,category)$ for depressed synapses, respectively.

## Topological metrics

For quantifying the non-random nature of changing connections, directed simplex counts were used. A $k$-simplex in a network $G$ is a set of $k + 1$ nodes (neurons) of $G$ that are all-to-all connected in a feedforward fashion. That is, there is a numbering of the nodes $v_0, v_1, \ldots v_k$, such that for every $i < j$ there is an edge from $v_i$ to $v_j$, and $k$ is called the dimension of the simplex. In particular, 0-simplices are the nodes of the network, 1-simplices directed edges (connections), and 2-simplices are also known as transitive triads. Random controls were defined as the same number of edges between the same nodes, resulting in the same 0- and 1-simplex counts. Given an edge in $G$, a notion of its edge-centrality in the network is its $k$-*edge indegree*, which is the number of $k$-simplices the edge is innervated by. Note that a $k$-simplex innervating an edge is equivalent to it being an edge in a $k + 2$-simplex going from the next to last to the last node of the simplex. This extends the classic notion of node indegree from nodes to edges. This value can be computed either for the simplices in the entire network or in a specified subnetwork e.g., the subnetwork on neurons belonging to an assembly.

Simplex counts and k-edge indegree values were computed using the analysis package (*Santander et al., 2024*), based on a fast C++ implementation from flagsercount (*Lütgehetmann et al., 2020*).

## MICrONS dataset

For the comparison of two of our findings with a rodent electron microscopy dataset with co-registered calcium-imaging traces, the *minnie65_public* release of the (*MICrONS, 2021*) dataset was used.

Synapses from sources other than one of the 60,048 classified neurons inside the bigger (*minnie65*) volume were discarded. The 117 version of the structural dataset was used for these analyses and made freely available in SONATA format (*Dai et al., 2020*) at https://doi.org/10.5281/zenodo.8364070. For comparable analysis, we restricted the volume to its central part ($650,000 \leq x \leq 950,000$ and $700,000 \leq z \leq 1,000,000$) and considered only E to E connections. Synapse volume was defined as the number of voxels painted by the automatic cleft segmentation (from the corresponding synapses_pni_2 table) times the $4 \times 4 \times 40$ nm voxel size. Total synapse volume was defined as the sum across all synapses mediating a connection.

For the functional dataset, eight calcium-imaging sessions from the 661 version were used. These sessions were selected by the same criteria as in *Reimann et al., 2024b*; *Santander et al., 2024*, namely at least 1000 neurons were scanned in each session and at least 85% of them were co-registered in the structural data. Neurons with non-unique identifiers were dropped, and the remaining ones restricted to the central part of the structural volume described above. Deconvolved spike trains of neurons (243 neurons/session on average, 1817 unique neurons in total, 2% and 15% of all cells, respectively) were used to calculate Pearson correlations. To average spike correlations from different sessions without any bias, they were first z-scored within each session.

### Spike time reliability

Spike time reliability was defined as the mean of the cosine similarities of a given neuron's mean-centered, smoothed spike times across all pairs of repetitions of a single pattern (*Schreiber et al., 2003*; *Cutts and Eglen, 2014*). To smooth the spike times, they were first binned to 1ms time bins, and then convolved with a Gaussian kernel with a standard deviation of 10ms as in *Ecker et al., 2024*; *Santander et al., 2024*.

### Visualization

Rendering of cells from the cortical network on *Figure 1A* was done with Brayns, while the rendering of selected cells with their thalamo-cortical synapses on D with BioExplorer. Morphologies on *Figures 1 and 6* were rendered with NeuroMorphoVis (*Abdellah et al., 2018*).

## Acknowledgements

The authors thank Nicolas Ninin for his involvement in the early stage of this project, Elvis Boci and Cyrille Favreau for their help with visualizations, Michael Gevaert, Joni Herttuainen, and Thomas Delemontex for their assistance with software engineering, and Alberto Antonietti, Christoph Pokorny, Kathryn B Hess, Ran Levi, Wulfram Gerstner, Roberto Araya, and Henry Markram for discussions. This study was supported by funding to the Blue Brain Project, a research center of the École polytechnique fédérale de Lausanne (EPFL), from the Swiss government's ETH Board of the Swiss Federal Institutes of Technology. EBM and DP were further supported by funding from the Institute for Data Valorization (IVADO), the CHU Sainte-Justine Research Center (CHUSJRC), Fonds de Recherche du Québec–Santé (FRQS), the Canada CIFAR AI Chairs Program, the Quebec Institute for Artificial Intelligence (Mila), and Google. Their compute infrastructure was supported through a grant from the Canada Foundation for Innovation (John Evans Leader Fund), and a grant of computing time awarded to from the Digital Research Alliance of Canada.

## Additional information

### Competing interests

Eilif B Muller: Reviewing editor, eLife. The other authors declare that no competing interests exist.

## Funding

| Funder | Grant reference number | Author |
|---|---|---|
| ETH Board | | András Ecker<br>Daniela Egas Santander<br>Marwan Abdellah<br>Jorge Blanco Alonso<br>Sirio Bolaños-Puchet<br>Giuseppe Chindemi<br>James B Isbister<br>James King<br>Pramod Kumbhar<br>Ioannis Magkanaris<br>Michael W Reimann |
| Institute for Data Valorization | | Dhuruva Priyan Gowri Mariyappan<br>Eilif B Muller |
| CHU Sainte-Justine Research Center | | Dhuruva Priyan Gowri Mariyappan<br>Eilif B Muller |
| Fonds de Recherche du Québec - Santé | | Dhuruva Priyan Gowri Mariyappan<br>Eilif B Muller |
| Canadian Institute for Advanced Research | AI Chairs Program | Dhuruva Priyan Gowri Mariyappan<br>Eilif B Muller |
| Quebec Institute for Artificial Intelligence (Mila) | | Dhuruva Priyan Gowri Mariyappan<br>Eilif B Muller |
| Google | | Dhuruva Priyan Gowri Mariyappan<br>Eilif B Muller |
| Canada Foundation for Innovation | John Evans Leader Fund | Dhuruva Priyan Gowri Mariyappan<br>Eilif B Muller |
| Digital Research Alliance of Canada | | Dhuruva Priyan Gowri Mariyappan<br>Eilif B Muller |

The funders had no role in study design, data collection and interpretation, or the decision to submit the work for publication.

## Author contributions

András Ecker, Conceptualization, Software, Formal analysis, Validation, Investigation, Visualization, Methodology, Writing – original draft, Writing – review and editing; Daniela Egas Santander, Conceptualization, Software, Formal analysis, Validation, Methodology, Writing – original draft, Writing – review and editing; Marwan Abdellah, Visualization; Jorge Blanco Alonso, Ioannis Magkanaris, Resources; Sirio Bolaños-Puchet, Software, Validation, Writing – review and editing; Giuseppe Chindemi, Conceptualization, Investigation, Methodology, Writing – review and editing; Dhuruva Priyan Gowri Mariyappan, Validation; James B Isbister, Validation, Methodology, Writing – review and editing; James King, Pramod Kumbhar, Software; Eilif B Muller, Conceptualization, Methodology, Writing – review and editing; Michael W Reimann, Conceptualization, Software, Formal analysis, Supervision, Visualization, Methodology, Writing – original draft, Project administration, Writing – review and editing

## Author ORCIDs

András Ecker ⓘ https://orcid.org/0000-0001-9635-4169
Sirio Bolaños-Puchet ⓘ https://orcid.org/0000-0003-4049-6488
James B Isbister ⓘ https://orcid.org/0000-0002-1013-3013
Eilif B Muller ⓘ https://orcid.org/0000-0003-4309-8266
Michael W Reimann ⓘ https://orcid.org/0000-0003-3455-2367

Reviewer #1 (Public review): https://doi.org/10.7554/eLife.101850.3.sa1
Reviewer #2 (Public review): https://doi.org/10.7554/eLife.101850.3.sa2
Reviewer #3 (Public review): https://doi.org/10.7554/eLife.101850.3.sa3
Author response https://doi.org/10.7554/eLife.101850.3.sa4

## Additional files

### Supplementary files
MDAR checklist

### Data availability
The 2.4 mm$^3$ subvolume of the juvenile rat somatosensory cortex, containing 211,712 neurons and 312,709,576 plastic synapses in SONATA format (*Dai et al., 2020*) is freely available at: https://doi.org/10.5281/zenodo.8158471. Exemplary Jupyter notebooks using the packages listed in the Key resources table above were deposited in the same repository on Zenodo.

The following dataset was generated:

| Author(s) | Year | Dataset title | Dataset URL | Database and Identifier |
| --- | --- | --- | --- | --- |
| Ecker A, Santander DE, Reimann MW | 2024 | Assemblies, synapse clustering and network topology interact with plasticity to explain structure-function relationships of the cortical connectome | https://doi.org/10.5281/zenodo.8158471 | Zenodo, 10.5281/zenodo.8158471 |

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
