## [Editor Report · eLife Assessment]

This **useful** study presents a biologically realistic, large-scale cortical model of the rat's non-barrel somatosensory cortex, investigating synaptic plasticity of excitatory connections under varying patterns of external activations and characterizing relations between network architecture and plasticity outcomes. The model offers an impressive level of biological detail, addressing many aspects of the cellular and network anatomy and properties, and investigating their relationships to the biologically plausible plasticity. The numerical simulations appear to be well executed and documented, providing an excellent resource to the community. The evidence supporting the main conclusions is **solid** with results being more observational in nature, and minor weaknesses relating to the lack of explanatory power of causal relationships and mechanisms.

---

## [Referee Report · Reviewer #1 (Public review)]

This paper investigates the dynamics of excitatory synaptic weights under a calcium-based plasticity rule, in long (up to 10 minutes) simulations of a 211,000-neuron biophysically detailed model of a rat cortical network.

Strengths

(1) A very detailed network model, with a large number of neurons, connections, synapses, etc., and with a huge number of biological considerations implemented in the model.

(2) A carefully developed calcium-based plasticity rule, which operates with biologically relevant variables like calcium concentration and NMDA conductances.

(3) The study itself is detailed and thorough, covering many aspects of the cellular and network anatomy and properties and investigating their relationships to plasticity.

(4) The model remains stable over long periods of simulations, with the plasticity rule maintaining reasonable synaptic weights and not pushing the network to extremes.

(5) The variety of insights the authors derive in terms of relationships between the cellular and network properties and dynamics of the synaptic weights are potentially interesting for the field.

(6) Sharing the model and the associated methods and tools is a big plus.

Weaknesses

(1) Conceptually, there seems to be a missed opportunity here in that it is not clear what the network learns to do. The authors present 10 different input patterns, the network does some plasticity, which is then analyzed, but we do not know whether the learning resulted in anything functionally significant. Did the network learn to discriminate the patterns much better than at the beginning, to capture or anticipate the timing of pattern presentation, detect similarities between patterns, etc.? This is important to understand if one wants to assess the significance of synaptic changes due to plasticity. For example, if the network did not learn much new functionally, relative to its initial state, then the observed plasticity could be considered minor and possibly insufficient. In that case, were the network to learn something substantial, one would potentially observe much more extensive plasticity, and the results of the whole study could change, possibly including the stability of the network. While this could be a whole separate study, this issue is of central importance, and it is hard to judge the value of the results when we do not know what the network learned to do, if anything.

(2) In this study, plasticity occurs only at E-to-E connections but not at others. However, it is well known that inhibitory connections in the cortex exhibit at the very least a substantial short-term plasticity. One would expect that not including these phenomena would have substantial consequences on the results.

(3) Lines 134-135: "We calibrated layer-wise spontaneous firing rates and evoked activity to brief VPM inputs matching in vivo data from Reyes-Puerta et al. (2015)."

(4) Can the authors show these results? It is an important comparison, and so it would be great to see firing rates (ideally, their distributions) for all the cell types and layers vs. experimental data, for the evoked and spontaneous conditions.

(5) That being said, the Reyes-Puerta et al. paper reports firing rates for the barrel cortex, doesn't it? Whereas here, the authors are simulating a non-barrel cortex. Is such a comparison appropriate?

(6) Comparison with STDP on pages 5-7 and Figure 2: if I got this right, the authors applied STDP to already generated spikes, that is, did not run a simulation with STDP. That seems strange. The spikes they use here were generated by the system utilizing their calcium-based plasticity rule. Obviously, the spikes would be different if STDP was utilized instead. The traces of synaptic weights would then also be different. The comparison therefore is not quite appropriate, is it?

(7) Section 2.3 and Figure 5: I am not sure this analysis adds much. The main finding is that plasticity occurs more among cells in assemblies than among all cells. But isn't that expected given what was shown in the previous figures? Specifically, the authors showed that for cells that fire more, plasticity is more prominent. Obviously, cells that fire little or not at all won't belong to any assemblies. Therefore, we expect more plasticity in assemblies.

(8) Section 2.4 and Figure 6: It is not clear that the results truly support the formulation of the section's title ("Synapse clustering contributes to the emergence of cell assemblies, and facilitates plasticity across them") and some of the text in the section. What I can see is that the effect on rho is strong for non-clustered synapses (Figure 6C and Figure S8A). In some cases, it is substantially higher than what is seen for clustered synapses. Furthermore, the wording "synapse clustering contributes to the emergence of cell assemblies" suggests some kind of causal role of clustered synapses in determining which neurons form specific cell assemblies. I do not see how the data presented supports that. Overall, it appears that the story about clustered synapses is quite complicated, with both clustered and non-clustered synapses driving changes in rho across the board.

(9) Section 2.5 and Figure 7: Can we be certain that it is the edge participation that is a particularly good predictor of synaptic changes and/or strength, as opposed to something simpler? For example, could it be the overall number of synapses, excitatory synapses, or something along these lines, that the source and/or target neurons receive, that determine the rho dynamics? And then, I do not understand the claim that edge participation allows one to "delineate potentiation from depression". The only related data I can find is in Figure 7A3, about which the authors write "this effect was stronger for potentiation than depression". But I don't see what they mean. For both depression and facilitation, the changes observed are in the range of ~12% of probability values. And even if the effect is stronger, does it mean one can "delineate" potentiation from depression better? What does it mean, to "delineate"? If it is some kind of decoding based on the edge participation, then the authors did not show that.

(10) "test novel predictions in the MICrONS (2021) dataset, which while pushing the boundaries of big data neuroscience, was so far only analyzed with single cells in focus instead of the network as a whole (Ding et al., 2023; Wang et al., 2023)." That is incorrect. For example, the whole work of Ding et al. analyzes connectivity and its relation to the neuron's functional properties at the network level.

Comments on revisions:

The authors addressed all my concerns from the previous review, primarily via textual changes such as improved Discussion. Thus, most of the weaknesses raised in the original review are not eliminated - in particular, points 1, and 5-9 - but they are acknowledged and described better. This remains a useful study that should be of interest to researchers in the field.

---

## [Referee Report · Reviewer #2 (Public review)]

Summary:

This paper aims at understanding the effects of plasticity in shaping dynamics and structure of cortical circuits, as well as on how that depends on aspects as network structure and dendritic processing.

Strengths:

The level of biological detail included is impressive, and the numerical simulations appear to be well executed. Additionally, they have done a commendable job in open-sourcing the model.

Weaknesses (after revision):

- As noted in my initial review, the observation that network activity remains stable without an explicit homeostatic mechanism-while acknowledged by the authors as consistent with previous findings (e.g., Higgins et al., 2014)-is not clearly framed as a replication or validation step in the current manuscript. For instance, the abstract states: "In our exploratory simulations, plasticity acted sparsely and specifically, firing rates and weight distributions remained stable without additional homeostatic mechanisms," without noting that this outcome has been previously reported, albeit in models with different levels of biological detail. Furthermore, in the general response to reviewers, the authors list this as the first item in their summary of phenomena accounted for by the model, which gives the impression that it is being presented as a primary result.

If this finding is instead meant to serve as a necessary validation that prior results continue to hold under the authors' extended modeling framework-including multicompartmental neurons, stochastic synaptic transmission, and a modified calcium-based plasticity rule-this should be made more explicit in both the abstract and main text. Unless there were specific reasons to suspect that these model extensions might disrupt previously observed stability, the conceptual contribution of this validation step remains unclear.

I would encourage the authors to revise the manuscript to clarify the role and novelty of this result in the context of existing literature and to briefly motivate why confirming this property in their model was an important step.

- While the revised manuscript includes improvements in the discussion of the generality and specificity of the findings, it still offers limited interpretability and mechanistic insight. As it stands, the simulations provide limited understanding of the underlying principles or mechanisms at play, which constrains the broader conclusions that can be drawn from the work.

- In my first review, I suggested that the comparison with the MICrONS dataset could be made more informative-specifically by showing the same quantification of Figure 7D (7B in the previous version) in a version of the model without plasticity and clarifying the interpretation of Figure 8B, where the data appears to align closely with the model before plasticity.

In their response, the authors explain that several of these features remain largely unchanged before and after plasticity. For example, they note that total \begin{document}$g_{\text{AMPA}}$\end{document} increases with \begin{document}$k$\end{document}-edge indegree even in the initial model configuration. I appreciate this clarification, but it highlights a conceptual point that should be more clearly addressed in the manuscript. If the aspects of the model that align with MICrONS data are already present before plasticity, then these similarities reflect properties of the initial network architecture or baseline dynamics, rather than outcomes shaped by the plasticity process itself.

If this interpretation is correct, it represents an interesting and potentially important finding. However, it is not currently articulated in the text. The manuscript places strong emphasis on the role of plasticity in shaping network structure and dynamics, yet the comparisons with MICrONS data appear to reflect features that do not depend on plasticity. Clarifying this distinction would help readers better appreciate the implications of the model-data comparison and discern which conclusions are genuinely supported by the data.

---

## [Referee Report · Reviewer #3 (Public review)]

Summary:

Ecker et al. utilized a biologically realistic, large-scale cortical model of the rat's non-barrel somatosensory cortex, incorporating a calcium-dependent plasticity rule to examine how various factors influence synaptic plasticity under in vivo-like conditions. Their analysis characterized the resulting plastic changes and revealed that key factors, including the co-firing of stimulus-evoked neuronal ensembles, the spatial organization of synaptic clusters, and the overall network topology, play an important role in affecting the extent of synaptic plasticity.

Strengths:

The detailed, large-scale model employed in this study enables the evaluation of diverse factors across various levels that influence the extent of plastic changes. Specifically, it facilitates the assessment of synaptic organization at the subcellular level, network topology at the macroscopic level, and the co-activation of neuronal ensembles at the activity level. Moreover, modeling plasticity under in vivo-like conditions enhances the model's relevance to experiments.

Weaknesses:

The paper lacks mechanistic insights into the observed phenomena, particularly regarding aspects that are typically inaccessible in traditional simplified models, such as layer-specific and layer-to-layer pathway-specific plasticity changes.

---

## [Author Response]

The following is the authors’ response to the original reviews

General response

Our modeling study integrates recent experimental advances on dendritic physiology, biophysical plasticity rules, and network connectivity motifs into a single model, aiming to clarify their hypothesized inseparable functional roles in neocortical learning. By modelling excitatory plasticity in multi-synaptic connections on dendrites within a network with biologically constrained higher-order structure, we show these aspects are sufficient to account for a wide range of interesting phenomena: First, the calcium-based plasticity rule acted sparsely and specifically, keeping the network stable without requiring homeostatic mechanisms or inhibitory plasticity, as usually employed for models based on STDP rules. Most importantly, simulations of the network initiated in a recurrent-excitation induced synchronous state transitioned to an in vivo-like asynchronous state, and remained there. Second, plastic changes were stimulus-dependent and could be predicted by neurons’ membership in functional assemblies, spatial clustering of synapses on dendrites, and the topology of the network’s connectivity. Several of our predictions could be confirmed by comparison to the MICrONS dataset.

Our study thus aims to provide a first broad exploration of these phenomena and their interactions in a model, as well as a foundation for future studies that examine specific aspects more deeply. Specific concerns of the reviewers about parameter choices (reviewer 2’s 2nd point - 2.2), claims about stability (2.1 and 3.1), the STDP control (1.5), and the motivation behind network metrics (1.8, 2.3) are addressed in detail below and in the revised manuscript.

**Reviewer #1 (Public review):**
This paper investigates the dynamics of excitatory synaptic weights under a calcium-based plasticity rule, in long (up to 10 minutes) simulations of a 211,000-neuron biophysically detailed model of a rat cortical network.Strengths(1) A very detailed network model, with a large number of neurons, connections, synapses, etc., and with a huge number of biological considerations implemented in the model.(2) A carefully developed calcium-based plasticity rule, which operates with biologically relevant variables like calcium concentration and NMDA conductances.(3) The study itself is detailed and thorough, covering many aspects of the cellular and network anatomy and properties and investigating their relationships to plasticity.(4) The model remains stable over long periods of simulations, with the plasticity rule maintaining reasonable synaptic weights and not pushing the network to extremes.(5) The variety of insights the authors derive in terms of relationships between the cellular and network properties and dynamics of the synaptic weights are potentially interesting for the field.(6) Sharing the model and the associated methods and tools is a big plus.

We thank the reviewer for their comments.

Weaknesses(1) Conceptually, there seems to be a missed opportunity here in that it is not clear what the network learns to do. The authors present 10 different input patterns, the network does some plasticity, which is then analyzed, but we do not know whether the learning resulted in anything functionally significant. Did the network learn to discriminate the patterns much better than at the beginning, to capture or anticipate the timing of pattern presentation, detect similarities between patterns, etc.? This is important to understand if one wants to assess the significance of synaptic changes due to plasticity. For example, if the network did not learn much new functionally, relative to its initial state, then the observed plasticity could be considered minor and possibly insufficient. In that case, were the network to learn something substantial, one would potentially observe much more extensive plasticity, and the results of the whole study could change, possibly including the stability of the network. While this could be a whole separate study, this issue is of central importance, and it is hard to judge the value of the results when we do not know what the network learned to do, if anything.

(1.1) The reviewer raises a very interesting point of discussion. As they remarked, it is very hard to judge what the network learned to do. However, our model was not designed to solve a specific task and even defining precisely what "learning" entails in a primary sensory region is still an open question. As many before us, we hypothesized that one of the roles of the primary somatosensory cortex would be to represent stimuli features and that most of the learning process would happen in an unsupervised manner. This is indeed what we have demonstrated by showing the stimulus-specificity of changes as well as an increase of reliability of assembly sequences between repetitions after plasticity. We have added this to the Discussion in lines 523-525.

(2) In this study, plasticity occurs only at E-to-E connections but not at others. However, it is well known that inhibitory connections in the cortex exhibit at the very least a substantial short-term plasticity. One would expect that not including these phenomena would have substantial consequences on the results.

(1.2) This is indeed well known. Please consider that we do have short-term plasticity (called synapse dynamics in the manuscript) at all connections, including inhibitory ones. We thank the reviewer for pointing out this potential confusion in the wording. We have now clarified this in the Methods in lines: 691-697. Furthermore, we have listed not having long-term plasticity at inhibitory connections in the limitations part of the Discussion in line: 593.

(3) Lines 134-135: "We calibrated layer-wise spontaneous firing rates and evoked activity to brief VPM inputs matching in vivo data from Reyes-Puerta et al. (2015)."(4) Can the authors show these results? It is an important comparison, and so it would be great to see firing rates (ideally, their distributions) for all the cell types and layers vs. experimental data, for the evoked and spontaneous conditions.

(1.3) The layer- and cell type specific spontaneous firing rates were indeed hidden in the Methods and on Supplementary Figure S3. We now reference that figure in the Results in line: 136. Furthermore, we have amended Supplementary Figure S3 (panel A2), to show these rates in the evoked state as well.

(5) That being said, the Reyes-Puerta et al. paper reports firing rates for the barrel cortex, doesn't it? Whereas here, the authors are simulating a non-barrel cortex. Is such a comparison appropriate?

(1.4) As correctly pointed out by the reviewer, we made the assumption that these rates would generalize to the whole S1 because of the sparsity of experimental data. This assumption is discussed in length in Isbister et al. (2023) and now in the limitations part of the Discussion in lines: 564-568.

(6) Comparison with STDP on pages 5-7 and Figure 2: if I got this right, the authors applied STDP to already generated spikes, that is, did not run a simulation with STDP. That seems strange. The spikes they use here were generated by the system utilizing their calcium-based plasticity rule. Obviously, the spikes would be different if STDP was utilized instead. The traces of synaptic weights would then also be different. The comparison therefore is not quite appropriate, is it?

(1.5) Yes, the reviewer's understanding is correct. However, considering the findings of Morrison et al. 2007 [PMID: 17444756], and Zenke et al. 2017 [PMID: 28431369] (cited in the manuscript in lines: 165-166), running STDP in a closed loop simulation would most likely make the network “blow up” because of the positive feedback loop. Thus, we argue that our comparison is more conservative, since by using pre-generated spikes, we opened the loop and avoided positive feedback. This is now further explained in lines: 166-167.

(7) Section 2.3 and Figure 5: I am not sure this analysis adds much. The main finding is that plasticity occurs more among cells in assemblies than among all cells. But isn't that expected given what was shown in the previous figures? Specifically, the authors showed that for cells that fire more, plasticity is more prominent. Obviously, cells that fire little or not at all won't belong to any assemblies. Therefore, we expect more plasticity in assemblies.

(1.6) We thank the reviewer for this comment. We added additional panels (G1 and G2) to Figure 5 (and describe their content in lines: 329-337) showing that this is not the case. Firing-rate alone is indeed predictive of plastic changes, but co-firing in assemblies is even more so.

(8) Section 2.4 and Figure 6: It is not clear that the results truly support the formulation of the section's title ("Synapse clustering contributes to the emergence of cell assemblies, and facilitates plasticity across them") and some of the text in the section. What I can see is that the effect on rho is strong for non-clustered synapses (Figure 6C and Figure S8A). In some cases, it is substantially higher than what is seen for clustered synapses. Furthermore, the wording "synapse clustering contributes to the emergence of cell assemblies" suggests some kind of causal role of clustered synapses in determining which neurons form specific cell assemblies. I do not see how the data presented supports that. Overall, it appears that the story about clustered synapses is quite complicated, with both clustered and non-clustered synapses driving changes in rho across the board.

(1.7) We agree with the reviewer, it is “quite complicated” and we also see that the writing could have been better/more precise and supported by the data shown on the Figure. We updated both the section title and a big chunk of the text to take the suggestions into account in lines: 361-373.

(9) Section 2.5 and Figure 7: Can we be certain that it is the edge participation that is a particularly good predictor of synaptic changes and/or strength, as opposed to something simpler? For example, could it be the overall number of synapses, excitatory synapses, or something along these lines, that the source and/or target neurons receive, that determine the rho dynamics? And then, I do not understand the claim that edge participation allows one to "delineate potentiation from depression". The only related data I can find is in Figure 7A3, about which the authors write "this effect was stronger for potentiation than depression". But I don't see what they mean. For both depression and facilitation, the changes observed are in the range of ~12% of probability values. And even if the effect is stronger, does it mean one can "delineate" potentiation from depression better? What does it mean, to "delineate"? If it is some kind of decoding based on the edge participation, then the authors did not show that.

(1.8) We thank the reviewer for this comment. We have included an analysis of the predictive power of indegree of the pre and postsynaptic neuron of a connection on the rho dynamics in Figure 7 (panel B). Please consider, that the rho dynamics are described on the level of connections, while properties like indegree are on the level of nodes. Any procedure transferring a node based property to an edge based property involves choices e.g., should the values be added, multiplied, should one be preferential over the other, or should they be considered independently? As edge-based metrics avoid these arbitrary choices, we would argue that they are - ultimately - the simpler and more natural choice in this context.

Though we believe that the metric of edge participation is simple, we recognize it is perhaps not common. Thus, we have switched to using a version of it that is perhaps more intuitive for the community at large i.e., as a metric of common innervation. Moreover, we have changed the name “(k+2) edge participation” to “(k)-edge indegree”, to make it even more accessible. For k=0, this is the number of neurons that commonly innervate the connection, i.e., a common neighbour. And for k=1, this is the number of connections that commonly innervate the connection. This is equivalent to edge participation from the next to last to the last neuron in a simplex. Furthermore, in lines: 391-418 we have added additional text and references explaining the intuition of why we think this metric is relevant, as it has been shown to affect correlated activity of pairs of neurons, as well as assembly formation.

Furthermore, we have clarified the language referring to potentiation and depression in lines: 420-422 and 448.

(10) "test novel predictions in the MICrONS (2021) dataset, which while pushing the boundaries of big data neuroscience, was so far only analyzed with single cells in focus instead of the network as a whole (Ding et al., 2023; Wang et al., 2023)." That is incorrect. For example, the whole work of Ding et al. analyzes connectivity and its relation to the neuron's functional properties at the network level.

(1.9) We thank the reviewer for pointing this out. Indeed, the sentence was improperly worded. We have appropriately changed this phrasing in lines: 616-618.

**Reviewer #2 (Public review):**
Summary:This paper aims to understand the effects of plasticity in shaping the dynamics and structure of cortical circuits, as well as how that depends on aspects such as network structure and dendritic processing.Strengths:The level of biological detail included is impressive, and the numerical simulations appear to be well executed. Additionally, they have done a commendable job in open-sourcing the model.

We thank the reviewer for their comments.

Weaknesses:The main result of this work is that activity in their network model remains stable without the need for a homeostatic mechanism. However, as the authors acknowledge, this has been demonstrated in previous studies (e.g., Higgins et al. 2014). In those studies, stability was attributed to calcium-based rules combined with calcium concentrations at in vivo levels and background neuronal activity. Since the authors use the same calcium-based rule, it is unclear what new result, if any, is being presented. If the authors are suggesting that the mechanism in their simulations differs, that should be stated clearly, and evidence supporting that claim should be provided.

(2.1) We do not see this as the main result of our study, but rather a critical validation step, since our calcium rule, while similar to previous ones, is not exactly the same (see equations (1) and especially (2) in Methods). This has been clarified in the text in lines: 150-151. Note in particular, that one of the main differences is the stochastic synaptic transmission and the role of calcium concentration on the release probability. Furthermore, our model involves multicompartmental neurons instead of point neuron models, which to our knowledge was never tested before with calcium-based plasticity rules at the network level. Moreover, determining the time required for stability to be reached is a necessary step to set up the simulation parameters to test the main hypotheses about rules governing the plastic changes.

The other findings discussed in the paper are related to a characterization of the dependency of plastic changes on network structure. While this analysis is potentially interesting, it has the following limitations.First, I believe the authors should include an analysis of the generality and specificity of their results. All the findings seem to be derived from a single run of the simulation. How do the results vary with different network initializations, simulation times, or parameter choices?

(2.2) All simulations were run with 3 different random seeds (mentioned in the Methods) and now shown in Supplementary Figure S8 for some selected analyses. The maximum duration of our simulations were limited by our hardware constraints. However, from the long (10 minutes) simulation we concluded that most changes happen within the first minute. This is how we determined 2 minutes as the simulation time for all other experiments. Parameters determining both the spontaneous and evoked network state are discussed in length in Isbister et al. (2023) and while we acknowledge that they are only shown in Supplementary Figure S3, we did not want to lengthen the manuscript with redundant details but rather refer to reader to the manuscript where this is discussed at large.

Crucially, we tried slightly different parameters of the plasticity model in the early phases of the research, and while they changed the exact numerical values of our results, the main trends (i.e., stabilization time, assemblies, synapse clustering, and network topology influencing plastic changes) remained unchanged. This is now shown in Supplementary Figure S13 and referenced in the Discussion in lines: 572-575.

Second, the presentation of the results is difficult to follow. The characterization comes across as a long list of experiments, making it hard to identify a central message or distinguish key findings from minor details. The authors provide little intuition about why certain outcomes arise, and the complexity of the simulation makes it challenging - if not impossible - to determine which model elements are essential for specific results and which mechanisms drive emergent properties. Additionally, the text often lacks crucial details. For instance, the description of k-edge participation should be expanded, and an explanation of what this method quantifies should be included. Overall, I believe the authors should focus on a smaller set of significant results and provide a more in-depth discussion.

(2.3) We acknowledge the complexity of these large-scale simulations and the interpretation of their results. We appreciate the reviewer's feedback on the areas that needed more detail. To address this, we have extended the Results section describing k-edge indegree with more background and intuition in lines: 391-418. See also our reply to reviewer 1 (1.8) above.

While the manuscript may appear to be "a long list of experiments," it is actually guided by the following logic: We choose a calcium-based rule because it was the natural choice in a multicompartmental model which already included calcium dynamics and NMDA receptors. After setting up the main network state, verifying stability (Figure 2), doing traditional basic analysis (Figure 3), and verifying that the changes are non-random (Figure 4); we elaborated on long-standing ideas about co-firing in cell assemblies (Figure 5) and spatial clustering of synapse on dendrites (Figure 6) interacting with plasticity. Finally as we had access to the network’s non-random connectivity we tried to link the network's topology to the observed plastic changes. This was done with a higher order perspective, given that there was previous evidence for the relevance of these structures on cofiring and correlated activity.

While we understand the frustration, we would highlight that the study is the first of its kind at this scale and level of biological detail. Our goal was to offer a broad exploration of the factors influencing plasticity and their interactions at this scale. Thus, laying the groundwork for future studies to investigate specific aspects more deeply.

The comparison of the model with the MICrONS dataset could be improved. In Figure 7B, the authors should show how the same quantification looks in a network model without plasticity. In Figure 8B, the data aligns with the model before plasticity, so it's unclear how this serves as a verification of the theoretical predictions.

(2.4) Our only claim is that by being used to working with both functional and structural data we were able to develop a metric (k-edge indegree) that could be utilized to study the non-random, high-order topology of the MICrONS connectivity as well. On Figure 8, spike correlations in MICrONS more or less align with both cases (before vs. after plasticity); the only difference is that spike correlations looked different enough in the model so we thought they are worth showing for both cases. Moreover, as the changes are sparse (Figure 2 and 3) the synapse strength panel of Figure 7(D) looks almost exactly the same before plasticity (see first two panels of Author response image 1). In line with our results, the small and significant changes increase as k-edge indegree increases (last panel of Author response image 1). As the first two panels look almost the same and the third one is shown in a slightly different way (Figure 7C2) we would prefer not to include this in the manuscript, but only in our response.

**Author response image 1. sa4fig1:** 

**Reviewer #3 (Public review):**
Summary:Ecker et al. utilized a biologically realistic, large-scale cortical model of the rat's non-barrel somatosensory cortex, incorporating a calcium-dependent plasticity rule to examine how various factors influence synaptic plasticity under in vivo-like conditions. Their analysis characterized the resulting plastic changes and revealed that key factors, including the co-firing of stimulus-evoked neuronal ensembles, the spatial organization of synaptic clusters, and the overall network topology, play an important role in affecting the extent of synaptic plasticity.Strengths:The detailed, large-scale model employed in this study enables the evaluation of diverse factors across various levels that influence the extent of plastic changes. Specifically, it facilitates the assessment of synaptic organization at the subcellular level, network topology at the macroscopic level, and the co-activation of neuronal ensembles at the activity level. Moreover, modeling plasticity under in vivo-like conditions enhances the model's relevance to experiments.

We thank the reviewer for their comments.

Weaknesses:(1) The authors claimed that, under in vivo-like conditions and in the presence of plasticity, firing rates and weight distributions remain stable without additional homeostatic mechanisms during a 10-minute stimulation period. However, the weights do not reach the steady state immediately after the 10-minute stimulation. Therefore, extended simulations are necessary to substantiate the claim.

(3.1) We thank the reviewer for this comment, as it gave us the opportunity to clarify in the text our stabilization criteria. Indeed, the dynamical system of weight changes has not reached a zero-change steady state because the changes, while small, are non-zero. However, in a stochastic system with ongoing activity (stimulus- or noise-driven), non-zero changes are expected. Thus, we consider the system to be at steady state when changes become negligible relative to a null model given by a random walk. Our results show that this condition is met around the 2-minute mark, with negligible changes in the subsequent 8 minutes.

Moreover, for spontaneous activity, we showed that an unstable network exhibiting synchronous activity can be stabilized into an asynchronous regime by the calcium-based plasticity rule within 10 minutes. These results show that the system reaches a stochastic steady state within 10 minutes without requiring homeostatic mechanisms. Our work reveals that incorporating more biological detail (i.e. calcium-based plasticity), reduces the need for additional mechanisms to stabilize network activity (e.g. fast homeostatic mechanisms).

Interestingly, one might argue that after 10 minutes of stimulation the network might transition to a different weight configuration if the stimuli change or cease. We agree this is an intriguing question, which we added to the Discussion in lines 611-613. However, this scenario concerns continuous learning, not the system’s steady-state dynamics.

(2) Another major limitation of the paper lies in its lack of mechanistic insights into the observed phenomena (particularly on aspects that are typically impossible to assess in traditional simplified models, like layer-specific and layer-to-layer pathways-specific plasticity changes), as well as the absence of discussions on the potential computational implications of the corresponding observed plastic changes.

(3.2) Our study integrates recent experimental advances aiming to clarify their hypothesized inseparable functional roles in neocortical learning. In particular, we study three different kinds of mechanistic insight: co-firing in assemblies (Figure 5), synapse clustering on postsynaptic dendrites (Figure 6), and high-order network topology (Figure 7). Furthermore, layer specificity is shown (Figure 3A1, B1, B2, D1) and so is layer-to-layer specificity (Figure 4A2). In addition we also describe synapse clustering on postsynaptic dendrites (Figure 6) which is not available in simplified models either.

As such, the mechanistic insights provided in our work are integrative in nature and aim to provide a first broad exploration of these phenomena and their interactions-which are rarely considered together in experimental or modelling studies. This foundation paves the way for future studies that examine specific aspects more deeply in this level of biological detail.

**Reviewer #1 (Recommendations for the authors):**
(1) I would suggest the authors explain more explicitly that their study uses plasticity for E-to-E connections and not others. Doing so in multiple places in the paper, but certainly in Methods and early in Results, would be helpful. This is stated in lines 117-119 ("To simulate long-term plasticity, we integrated our recently published calcium-based plasticity model that was used to describe functional long-term potentiation and depression between pairs of pyramidal cells"), but could be highlighted more.

We have added it to several lines in the Methods: 621, 648, 649.

(2) "Simulations were always repeated at least three times to assess the consistency of the results." This sounds important. How is this used for the analysis? Do the results reported combine the data from the 3 simulations? How did the authors check the "consistency of the results"? Did they run any statistical tests comparing the results between the 3 simulations or was it more of a visual check?

The reported results come from a single simulation. Three simulations were run to check that no obvious qualitative differences could be found, such as a change of network regime, association between stimuli and assemblies. No statistical tests can be run with samples of size three. These are now shown in Supplementary Figure S8, and additional clarifying text has been added in Methods line: 722.

(3) "We needed 12M core hours to run the simulation presented in this manuscript." The Methods section mentions ~2.4 M core hours for a 10-minute simulation, which may be confusing. It might be helpful to provide a table with all the simulations run for this study.

We wanted to provide a rough estimate of the runtime, but did not run a deep profiling of all campaigns. The results depend on the actual hardware and configurations used (e.g., temporal resolution of synapse reporting). We understand the potential source of confusion and have clarified this in the Methods in lines 719-721 (and took it out from the Discussion).

**Reviewer #2 (Recommendations for the authors):**
(1) I found the paper somewhat challenging to follow, as there are many small points, making it unclear what the main message is. It sometimes feels like a list of 'we did this and found that.' It might be helpful if the authors focused on a smaller number of key results with more in-depth discussion. For instance, the discussion of network topology on page 9 is intriguing but condensed into a single, dense paragraph that is hard to follow. Clarifying how the random control is generated would also be beneficial.

See our response to the public review’s third point (2.3).

(2) Line 245: typo? "Furthermore, the maximal simplex dimension found in the subgraph was two higher than expected by chance.".

We changed the grammar in line: 249.

(3) Line 410: typo? "It has been previously shown before that assemblies have many edges".

Noted and fixed in line: 463.

**Reviewer #3 (Recommendations for the authors):**
(1) The authors claimed that plasticity operates in a sparse and specific manner, with firing rates and weight distributions remaining stable without additional homeostatic mechanisms. However, as shown in Figure 2D inset, the weights do not reach their steady-state values immediately after the 10-minute stimulation. A similar issue is observed in Figure 2G. It would be necessary to show the claim is indeed true as the weights reach the steady states.

See our response to the public review’s first point (3.1).

(2) In the model, synapses undergo both short- and long-term plasticity, but the contribution of short-term plasticity to the stated claim is unclear. It would be helpful to demonstrate how the results of Figure 2 are affected when short-term plasticity is excluded.

STP is needed to achieve the asynchronous in vivo-like firing state in our model (and is intimately linked to the fitting procedure of the plasticity rules - mean-field approximation is not possible due to the important role of synaptic failures in thresholded plasticity outcomes), thus it cannot be excluded. We have added this to the Methods in lines: 691-697.

(3) It would be helpful to include a supplementary plot, similar to Figure 2F, illustrating the corresponding results for STDP.

This is not possible as we did not run a different simulation with STDP, only evaluated the changes in connections with an STDP model using spikes from our simulation. We did not incorporate the STDP equations into our detailed network, as there is no canonical or unambiguous way for doing so (e.g., one would need to handle the fact the connections are multi-synaptic). Note however, that considering the findings of Morrison et al. 2007 [PMID: 17444756], and Zenke et al. 2017 [PMID: 28431369] (cited in the manuscript in lines: 165-166), running STDP in a closed loop simulation would most likely make the network “blow up” because of the positive feedback loop.

(4) It would be helpful to provide mechanistic insights into the current observations and to discuss the potential computational implications of the observed plastic changes. Particularly on aspects that are typically impossible to examine in traditional models, like layer-specific plastic changes presented in Fig. 3A1, B1, B2, D1, and layer-to-layer pathways-specific plastic changes illustrated in Figure 4A2.

See our response to the public review’s second point (3.2).

(5) The use of the term 'assembly' in most places of the manuscript may cause confusion. To enhance clarity and foster effective discussions in the field, I would recommend replacing it with 'ensemble,' as suggested in Miehl et al. (2023), 'Formation and computational implications of assemblies in neural circuits' (The Journal of Physiology, 601(15), 3071-3090), which should also be cited.

We read the mentioned manuscript when it was published (and appreciated it a lot), now reference it, and explain why we did not exactly follow the suggestion in lines: 293-299.

(6) The title of Figure 5 is not directly supported by the current figure. To strengthen the alignment, it would be helpful to present the results from lines 303-306 in bar plots and incorporate them into Figure 5 to better substantiate the figure title.

While the mentioned lines compare maximum values to those within the whole dataset, we think those 2*12*12 values are better presented in condensed matrices than bar plots (while the maximum values are still easily grasped from the colorbars). We have added panel G2 to the figure to address a comment by reviewer 1 (1.7), we believe that this further supports the title of the Figure.

(7) Line 326, cite "Kirchner, J. H., & Gjorgjieva, J. (2021). Emergence of local and global synaptic organization on cortical dendrites. Nature Communications, 12(1), 4005." and "Kirchner, J. H., & Gjorgjieva, J. (2022). Emergence of synaptic organization and computation in dendrites. Neuroforum, 28(1), 21-30."

Although we were aware of the mentioned manuscripts, we did not include them originally because they are models of a different species. However, we have now cited these in line: 347.

(8) The contrast results for ensembles 11 and 12 do not appear to support the claims made in lines 339-341. Clarification on this point would be helpful.

The reviewer is right, we have updated lines: 360-361, to clarify the difference between the two late assemblies.

(9) For Figure 6C and 6D in Section 2.4, rather than presenting the results for individual ensembles (which could be moved to the supplementary materials), it would be easier if the authors could summarize the results by grouping them into three categories: early, middle, and late ensembles.

We agree with the reviewer’s suggestion and tried it before, but as the results slightly depend on functional assembly size as well (not only temporal order) averaging them loses information (see different xlims of the panels). Given that the issue is complex we decided to show all the data on the Figure, but we have revised the text now to provide a more high-level interpretation.